# CAN VISION-LANGUAGE MODELS ANSWER FACE TO FACE QUESTIONS IN THE REAL-WORLD?

**Reza Pourreza**[1,*]   **Rishit Dagli**[2,*,†]   **Apratim Bhattacharyya**[1]   **Sunny Panchal**[1]
**Guillaume Berger**[1]   **Roland Memisevic**[1]

[1]Qualcomm AI Research[‡]  [2]University of Toronto

[*]Equal contribution.

[†]Work completed during an internship at Qualcomm AI Research.

https://www.qualcomm.com/developer/software/qualcomm-interactive-video-dataset-qivd

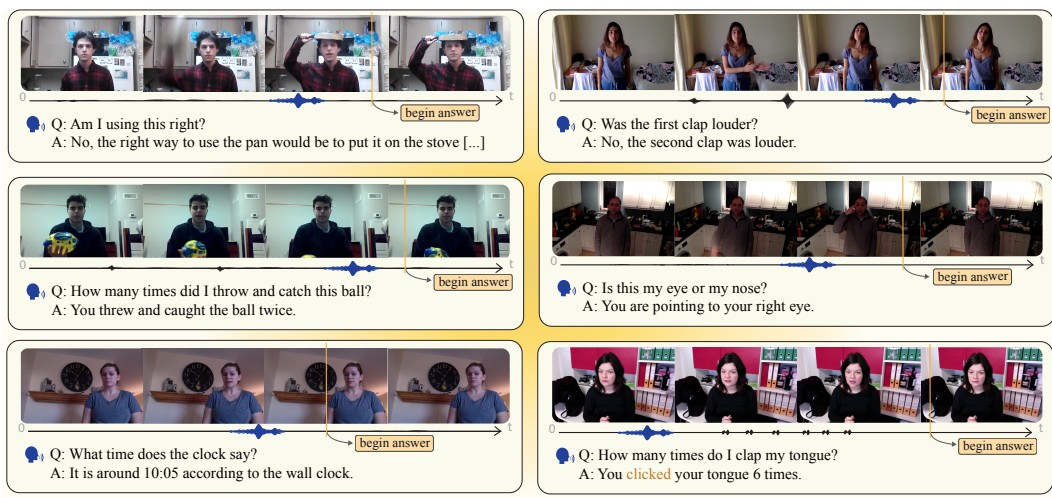

Figure 1: We present the Qualcomm Interactive Video Dataset (QIVD), a dataset collected in an online question-answering setup, where users pose open-ended questions using their camera and microphone. QIVD offers videos with raw audio, annotated textual transcriptions of the spoken questions, and text answers with annotated timestamps. These timestamps indicate when a question can be sensibly answered given the video context. QIVD serves as a realistic and challenging dataset for situated visual reasoning in Large Multi-modal Models.

## ABSTRACT

AI models have made significant strides in recent years in their ability to describe and answer questions about real-world images. They have also made progress in the ability to converse with users in real-time using audio input. This raises the question: have we reached the point where AI models, connected to a camera and microphone, can converse with users in real-time about scenes and events that are unfolding live in front of the camera? This has been a long-standing goal in AI and is a prerequisite for real-world AI assistants and humanoid robots to interact with humans in everyday situations. In this work, we introduce a new dataset and benchmark, the Qualcomm Interactive Video Dataset (QIVD), which allows us to assess the extent to which existing models can support these abilities, and to what degree these capabilities can be instilled through fine-tuning. The dataset is based on a simple question-answering setup, where users ask questions that the system has to answer, in real-time, based on the camera and audio input. We show that existing models fall far behind human performance on this task, and we identify the main sources for the performance gap. However, we also show that for many of the required perceptual skills, fine-tuning on this form of data can significantly reduce this gap.

---

[‡]Qualcomm AI Research is an initiative of Qualcomm Technologies, Inc.

# 1 INTRODUCTION

Recent advancements in Large Multimodal Models (LMM) have significantly enhanced the ability of AI systems to interact naturally and fluently with users in real-time. Existing AI agents can process audio, speech, and visual inputs to engage in conversations about images or videos. However, the conversational capabilities of state-of-the-art LMMs such as GPT-4o (Hurst et al., 2024) are limited to question-answering on visual understanding and reasoning tasks, such as describing images or answering questions that require inferring object positions and relations in the visual input. These systems often fail to provide truly situated, live, conversational experiences (Figure 1) that we may expect from humanoid robots or real-time video-call chatbots in the future.

We hypothesize that this limitation stems from the fact that current vision-language datasets and benchmarks are biased toward offline reasoning about images and videos. That is, the models receive the entire visual input and the entire question at once before being required to provide an answer. This is because the training data for such tasks can be easily sourced on the internet or easily generated through automated pipelines. There is a distinct lack of benchmarks and datasets that test genuine, real-time, "face-to-face" conversational skills. A separate but related problem is that models are not trained to respond at the appropriate time in a conversation – knowing "when to speak" is crucial for conducting real-world conversations, yet this timing skill remains underdeveloped and understudied in current benchmarks.

To address these challenges and assess the limitations of existing models, we introduce the Qualcomm Interactive Video Dataset (QIVD), a new dataset and benchmark designed for end-to-end trained systems aimed at real-time user interaction. QIVD is structured as an online question-answering setup, where users pose open-ended questions using their camera and microphone, and the system must respond appropriately. Our work differs fundamentally from other related datasets and benchmarks by introducing an entirely online question-answering paradigm where both questions and answers evolve in real-time as the video unfolds, requiring models to maintain contextual awareness while handling inherent ambiguities in human references to visual elements. We show how this simple type of interaction allows us to capture a rich set of visual concepts that fall under the umbrella of situated visual understanding, including deictic (referring) expressions, pointing gestures, object ambiguities, behavior and action understanding, and counting, as well as audio-visual concepts. An overview of our dataset is shown in Figure 1. Due to the in-the-wild nature of the recordings, the videos exhibit considerable variation in lighting conditions, background settings, the range and nature of questions posed, actions performed by subjects, and other audio-visual characteristics.

To showcase the unique challenges our dataset presents, we conduct a series of experiments where we evaluate multiple open and closed-source state-of-the-art models, and fine-tuned models on our dataset. Our experiments reveal that the seemingly simple interaction of answering questions live, in real-time, is highly challenging for existing AI systems (Hurst et al., 2024), even if they are otherwise good at performing visual reasoning. Our experiments indicate that the failure modes of existing systems can be attributed to their: (1) difficulty integrating visual and auditory information in real-time to disambiguate questions, (2) inability to determine the appropriate time at which to answer, and (3) inability to answer questions whose answers require situational common sense. Our dataset supports research on online LMMs capable of situated audio-visual reasoning and can be leveraged to build conversational agents that interact with users in real-time.

Our contributions are summarized as follows:

1. We introduce QIVD, a novel multi-modal dataset designed to evaluate online situated audio-visual reasoning and real-time conversational skills.[1]

2. We benchmark existing LMMs and identify critical weaknesses in their ability to handle real-life conversations.

3. We demonstrate that these limitations can be effectively mitigated by fine-tuning models on appropriate audio-visual conversational data.

4. We develop a simple yet effective baseline to process streaming audio-visual inputs, departing from traditional offline paradigms.

---

[1]qualcomm.com/developer/software/qualcomm-interactive-video-dataset-qivd

Table 1: Comparison of various benchmarks encompassing several key aspects.

| Benchmark | #Videos | #QA-Pairs | Annotation | Audio | Subtitle | Interactive | Face-to-Face |
|---|---|---|---|---|---|---|---|
| AVSD (DSTC7) (Alamri et al., 2018) | 11156 | ~111560 | Manual | ✓ | ✗ | ✓ | ✗ |
| KnowIT VQA (Garcia et al., 2020) | 207 | 24282 | Manual | ✓ | ✓ | ✗ | ✗ |
| LifeQA (Castro et al., 2020) | 275 | 2326 | Manual | ✓ | ✓ | ✗ | ✗ |
| How2QA (Li et al., 2020) | 9035 | 44007 | Manual | ✓ | ✓ | ✓ | ✗ |
| MedVidQA (Gupta et al., 2023) | 899 | 3010 | Manual | ✓ | ✓ | ✓ | ✗ |
| Social-IQ (Zadeh et al., 2019) | 1250 | 7500 | Manual | ✓ | ✗ | ✗ | ✓ |
| Video-MME (Fu et al., 2024) | 900 | 2700 | Manual | ✓ | ✓ | ✗ | ✗ |
| CodeVidQA (Raja et al., 2025) | 2104 | 2104 | Automatic | ✓ | ✓ | ✓ | ✗ |
| Ego4D Social Interactions (et. al., 2022) | 667 | task-specific *labels* | Manual | ✓ | ✗ | ✓ | ✓ |
| TVQA (Lei et al., 2018) | 21793 | 152545 | Manual | ✓ | ✓ | ✗ | ✗ |
| NExT-GQA (Xiao et al., 2024) | 1557 | 10531 | Manual | ✓ | ✓ | ✓ | ✗ |
| STAR (Wu et al., 2024) | 22000 | 60000 | Automatic | ✓ | ✗ | ✓ | ✗ |
| VStream-QA (Zhang et al., 2024) | 32 | 3500 | Automatic | ✓ | ✗ | ✓ | ✗ |
| QIVD | 2900 | 2900 | Manual | ✓ | ✓ | ✓ | ✓ |

## 2 RELATED WORK

**Offline Video Evaluation Benchmarks**: Prior work on video understanding benchmarks has primarily focused on offline evaluation paradigms. There have been multiple temporal video understanding benchmarks for open-domain understanding (Li et al., 2024b; Liu et al., 2024b; Xu et al., 2023; Patraucean et al., 2023; Ning et al., 2023; Hu et al., 2025), hand movements (Goyal et al., 2017; Materzynska et al., 2019), articulated motion (Dagli et al., 2024), full human body motion (Panchal et al., 2024), robotics (Haresh et al., 2024; Yu et al., 2024; Bao et al., 2023a; Gu et al., 2023; Brohan et al., 2023; Jiang et al., 2022), and embodied reasoning (Yang et al., 2025b). These benchmarks evaluate models' ability to comprehend temporal relationships but operate in a fully offline manner. Long-form video understanding has been addressed by datasets such as LVBench (Wang et al., 2025), and MoVQA (Zhang et al., 2023b), which extend the context window but fail to simulate real-time constraints. In contrast, our QIVD dataset and benchmark focuses on real-world QA.

**Situated Video Evaluation Benchmarks**: Situated question answering has also been studied by (Das et al., 2018; Ma et al., 2023) and follow-up works (*e.g.* (Wang et al., 2023; 2026; Wu et al., 2023)). A separate line of work has studied "common sense" situational understanding for AI models, albeit not in a VQA format. This includes the work by (Goyal et al., 2017; et. al., 2022; Patraucean et al., 2023) and recent work on situated live dialogue (Bao et al., 2023b; Panchal et al., 2024). Our work is similar in that it involves real-world interaction. In contrast to the existing work, questions in our dataset are free-form and open-ended rather than task-specific and oriented toward a specific goal.

In contrast to existing question-answering tasks, the task introduced in our work involves real-world interaction with a user, and as such the input is not confined to only visual information. Moreover, we place the task into a truly situated context, where correct answers require a true understanding of the scene unfolding in the real world. In contrast to that line of work, in this paper, we study situated question answering in a real-world not synthetic environment, by interacting "live" with a human subject, and by using audio and video input.

**Online Models**: Recent work on online video processing includes VideoLLM-online (Chen et al., 2024) and FlashVStream (Zhang et al., 2024), which attempt to address real-time processing constraints but remain limited in their ability to handle deictic references and situated understanding and also do not include audio. The StreamVLM (Panchal et al., 2024) supports situated understanding but is limited to the fitness domain and also lacks audio. Furthermore, existing benchmarks typically evaluate general visual understanding rather than modeling the situated, interactive nature of real-world human-AI conversations about visual content.

## 3 QIVD

The purpose of QIVD is to train and evaluate AI models on situated visual understanding. Each data instance comprises a video sequence annotated with temporally synchronized question-answer pairs. Additionally, the dataset includes the ground-truth answer to the question, making it possible to probe a model's understanding of the situation depicted in a given clip. Structuring the data as a simple

question-answering task allows us to separate situated understanding from multi-hop conversational capabilities. The latter is a similarly difficult but largely orthogonal challenge for existing models. A side-by-side comparison of the features offered by QIVD and other related datasets is presented in Table 1.

## 3.1 DATA COLLECTION

**Recording**: Crowd workers were instructed to record short videos using the camera and microphone of their mobile phone or laptop. They were free to choose the content of their videos but were shown examples featuring various gestures, actions, and objects to help them understand the dataset's purpose. The participants received written instructions explaining that these videos would be used to train and evaluate AI systems in understanding visual scenes. The instructions clarified that the AI system's purpose would be to correctly answer a single question rather than engage in a multi-step conversation. While recording their videos, crowd workers posed a question related to what was being shown. They were encouraged to be creative with their questions while ensuring they referenced the action or scene being recorded. After collection, all videos were inspected for audio and video quality, and their suitability for inclusion in the dataset.

**Annotation Methodology**: Each video in the QIVD dataset has three annotations. First, it includes a human-generated transcript of the question asked during the recording. Second, we provide a human-generated answer to that question. Third, we added a timestamp that marks the specific moment when it would be appropriate to answer the question. The timestamp does not always coincide with the end of the spoken question–in many cases, additional video context is required after the question was asked. For example, if a participant asked "What action is this?" *before* performing the action, the appropriate moment to answer would be *after* the action was visible in the video. This approach ensures that annotations also reflect when sufficient information becomes available to answer the question correctly, if required, rather than simply when the question ends. Finally, all of our submissions were reviewed by humans to verify their accuracy.

Unlike datasets constructed from pre-recorded videos with post-hoc annotations, our contemporaneous question-asking approach places a strong demand on situational context understanding. Our videos capture authentic uncertainty about future events in the video, including questions that genuinely test temporal reasoning, and require situational awareness to answer at the appropriate time. The annotations for answer timing are particularly valuable as they acknowledge that certain queries require monitoring the audio or visual stream over time to aggregate relevant information, and ascertaining when to respond. Through our collection approach, we provide a robust benchmark for evaluating a model's proficiency in understanding and responding to situated audio-visual stimuli. We show a few examples from our dataset in Figure 1 using four frames per video.

## 3.2 POST-PROCESSING WORKFLOW

Following the initial data collection phase, we perform comprehensive post-processing to enhance dataset utility by adding structured metadata and further ensure dataset quality. This section details our approach to quality assurance and taxonomic categorization of the dataset.

**Quality Checks**: To ensure data quality and ethical standards, we used a multi-stage quality control process. Each video underwent automated evaluation followed by manual inspection by trained evaluators who assessed the content according to predefined exclusion criteria. Specifically, we examined all videos for the presence of third persons, private data, and protected intellectual property; for the presence of inappropriate content, such as hate speech, and other potentially harmful elements; for linguistic compliance (clearly intelligible, English audio content); and for technical quality (absence of severe motion blur, compression artifacts, etc.).

After inspection, 2900 videos were deemed suitable and included in the dataset.

**Semantic Categorization**: To facilitate fine-grained analysis of model performance across different visual reasoning tasks, we developed a taxonomy of question types. The taxonomic structure allows for systematic evaluation of model performance across diverse visual reasoning tasks, enabling us to identify specific strengths and weaknesses in situated understanding capabilities. Each video-question pair was assigned to one or more of 13 predefined semantic categories representing distinct visual

Table 2: Dataset size metrics (total videos, vocabulary size), video characteristics (total frames, average length, frame rate, resolution), and linguistic properties of questions and answers. Average answer timestamps are represented by the average time in the video when the question should optimally be answered as a percentage of the video duration. The token statistics are calculated with the Llama-3 tokenizer. Standard deviations are shown in parentheses.

| Statistic | Value | Statistic | Value |
|---|---|---|---|
| Total Videos | 2900 | Avg. Answer Timestamp | 81.47% ($\pm$13.89) |
| Vocabulary Size (words) | 3624 | Avg. FPS | 30 ($\pm$0.00) |
| (tokens) | 3072 | | |
| Total Frames | 443350 | Question Types (Total) | |
| Avg. Video Length (s) | 5.10 ($\pm$0.44) | Questions with "where" | 47 |
| Avg. Question Length (words) | 6.09 ($\pm$1.94) | Questions with "how" | 512 |
| (tokens) | 7.60 ($\pm$2.28) | Questions with "what" | 1102 |
| Avg. Answer Length (words) | 7.23 ($\pm$4.31) | Deictic References (Total) | |
| (tokens) | 9.73 ($\pm$5.61) | Questions with "here" | 32 |
| Avg. Short Answer Length (words) | 1.38 ($\pm$0.82) | Questions with "these" | 39 |
| (tokens) | 1.98 ($\pm$1.27) | Questions with "that" | 45 |
| Avg. Resolution (width) | 640.00 ($\pm$0.00) | Questions with "there" | 105 |
| (height) | 382.29 ($\pm$46.01) | Questions with "this" | 568 |

Table 3: Distribution of samples across the 13 semantic categories in our dataset, with the answer timestamp as a percentage of video duration for each category. Percentages in the Samples column show the relative distribution of categories within the dataset.

| Category | Answer Timestamp | Samples | Category | Answer Timestamp | Samples |
|---|---|---|---|---|---|
| Action Attributes | 84.31% ($\pm$13.56) | 155 (5.34%) | Object Referencing | 79.18% ($\pm$13.61) | 706 (24.34%) |
| Action Counting | 92.22% ($\pm$8.73) | 225 (7.76%) | Object Understanding | 80.63% ($\pm$14.07) | 79 (2.72%) |
| Action Detection | 85.46% ($\pm$13.22) | 440 (15.17%) | Scene Understanding | 79.91% ($\pm$13.58) | 38 (1.31%) |
| Action Understanding | 81.47% ($\pm$15.07) | 110 (3.79%) | Audio-Visual | 90.09% ($\pm$11.49) | 22 (0.76%) |
| Object Attributes | 79.52% ($\pm$13.41) | 562 (19.38%) | OCR | 83.04% ($\pm$13.08) | 23 (0.79%) |
| Object Counting | 78.41% ($\pm$12.75) | 286 (9.86%) | Subjective | 77.39% ($\pm$15.15) | 43 (1.48%) |
| Object Detection | 76.95% ($\pm$15.65) | 211 (7.28%) | Total | 81.47% ($\pm$13.89) | 2900 (100%) |

reasoning capabilities. The categorization process uses a semi-automated approach: first, a large language model (LLM) is used to perform preliminary classification based on question content and transcribed answers; next, human annotators verify and refine the categories. Our semantic taxonomy encompasses the categories listed in Appendix B.2.

**Answer Normalization**: To facilitate better quantitative evaluation and reduce ambiguity in model assessment, we implemented an answer normalization process. For each original free-form response, we generated a condensed "short-answer" version that retained only the essential information required to correctly address the question. We follow a similar semi-automated method as semantic categorization for generating short answers. During evaluation, we use both the short answer and the original ground truth to evaluate models.

## 3.3 DATASET STATISTICS

**Dataset Composition**: The QIVD dataset consists of 2900 video clips and thus 2900 unique question-answer pairs. Table 2 summarizes the statistics of the dataset. The majority of clips have a length between 4 and 8 seconds. This range captures the natural timeframe in which a situated question about the visual scene can be posed and answered. We show the breakdown by the semantic taxonomy (Section 3.2) of the question-answer pairs in Table 3.

**Temporal Characteristics**: A distinctive feature of the QIVD dataset is the temporal relationship between the point in time when a question is posed and the point in time when sufficient information is available to answer it. We analyze the temporal characteristics by category in Table 3, which shows the distribution of optimal answer timestamps relative to video duration for each category. As we observe from Table 3, action-related categories generally require observing a larger portion of the video before answering, with Action Counting showing the highest optimal time (92.22% of video duration). This reflects the natural temporal dependency in action-related questions, where the answer often depends on observing the completion of an action sequence. In contrast, Object Detection

Table 4: **ASR performance comparison.** Evaluation of Automatic Speech Recognition (ASR) systems on the QIVD dataset using standard text similarity metrics. Timestamp detection accuracy is measured by $\Delta t$ that represents the mean absolute error in the optimal time to answer.

| Model | METEOR ↑ | BLEU ↑ | ROUGE-L ↑ | $\Delta t \downarrow$ | $\Delta t\,(-) \downarrow$ | $\Delta t\,(+) \downarrow$ |
|---|---|---|---|---|---|---|
| Whisper (Radford et al., 2023) | 90.01 | 80.95 | 90.32 | - | - | - |
| Whisper-Streaming (Machácek et al., 2023) | 92.34 | 74.57 | 91.82 | 0.83 | -0.94 | 0.61 |
| Stream-Qwen-Omni | - | - | - | 0.52 | -0.62 | 0.53 |

(76.95% of video duration) and Subjective questions (77.39% of video duration) can typically be answered earlier in the video, often right after the question is asked.

## 4 BASELINE STREAMING APPROACH

Two critical features of QIVD are:

**Self-Contained Videos**: The videos are self-contained, with the question embedded in the audio channel. An optimal model should be capable of answering these questions directly from the videos without the need for transcription.

**When-to-Answer Desiderata**: The videos are sufficiently long to include a scenario, a question, and any additional frames. An effective streaming model should identify the ideal moment to start answering the question, which is when both the question and any information necessary to answer it are present.

Current state-of-the-art LMMs do not integrate streaming and concurrent processing of audio and video information for situational interaction. To address this gap, we propose a novel streaming approach that pairs a streaming automatic speech recognition (ASR) system, used to transcribe questions and detect answer moments, with a Video-LMM to analyze video content and provide answers.

In detail, our streaming approach relies on the Streaming-Whisper model (Machácek et al., 2023) to identify "when to answer". The Streaming-Whisper model (Machácek et al., 2023) uses the LocalAgreement algorithm (Liu et al., 2020) to transcribe text in a streaming setup. The LocalAgreement algorithm transcribes the input audio in chunks and a subset of previous chunks are used to condition the transcription of the next chunk. In practice, we found that a chunk size of 0.25 seconds is sufficient for accurate streaming transcription for this dataset. Processing the input audio in chunks allows us to detect the end of the question asked by the participant in the video. It is important to note that, as mentioned above, the end of a question does not necessarily capture the optimal moment for an answer, as some necessary information may arise later in the video. Thus, we consider this approach as a reasonable compromise given the current limitations of ASR solutions and LMMs. After detecting the end of the question, the input video and audio up to that timestamp, along with the transcribed question, are provided as input to the LMM backbone. The LMM backbone can then process the multi-modal video and audio inputs along with the transcribed question to provide an answer. We explore different LMM backbones as outlined in Section 5.1.

## 5 EXPERIMENTS

We conduct comprehensive experiments to evaluate various open- and closed-source models on QIVD.

### 5.1 EXPERIMENTAL SETUP

**Configurations**: The experiments are performed within four distinct setups:

1. Streaming setup: Under this setup, we evaluate the baseline streaming approach introduced in Section 4.

2. Offline setup: In the baseline streaming approach, evaluating LMMs can be challenging due to potential inaccuracies in the questions extracted by the streaming ASR system, leading to accumulated errors. Therefore, in the offline setting, we use ground-truth questions to evaluate the models. This approach ensures that the evaluation is based on perfectly transcribed questions, allowing for an effective assessment of merely a model's answering performance. Consequently, the performance is an optimistic estimate of overall real-world performance.

3. Impact of audio: Among existing LMMs, the VideoLLaMA family of models (Zhang et al., 2023a) are state-of-the-art models capable of simultaneously processing both audio and video content. Although these models cannot transcribe speech, they can utilize audio content as a complementary source of information, thereby potentially enhancing accuracy. We evaluate these models by examining the impact of additional audio on the accuracy of their question-answering capabilities.

4. Impact of when-to-answer: This experiment investigates how the timing of the when-to-answer moment affects model performance. We utilize the Qwen2.5-Omni model (Xu et al., 2025), the only publicly available model capable of concurrently processing both audio and video modalities while also transcribing speech. The model is provided with both the ground-truth and ASR-derived when-to-answer timestamps. It then transcribes the question and generates an answer, allowing us to compare the outputs and assess the influence of timing on response quality.

**Baseline Models**: We experiment with various open-source and closed-source LMMs.

The open-source models we evaluate include InstructBLIP (7B) (Dai et al., 2023), Video-ChatGPT (7B) (Maaz et al., 2024), VideoChat (7B) (Li et al., 2024a), VideoChat2 (7B) (Li et al., 2024b), LLaVA-NeXT (7B) (Liu et al., 2024a), LLaMA-VID (13B) (Li et al., 2024c), Video-LLaMA (13B) (Zhang et al., 2023a), VideoLLaMA2 (7B/72B) (Cheng et al., 2024), VideoLLaMA2.1 (7B) (Cheng et al., 2024), VideoLLaMA3 (7B) (Zhang et al., 2025), Video-LLaVA (7B) (Zhu et al., 2023; Lin et al., 2023), VideoLLM-Online (Chen et al., 2024), Flash-VStream (Zhang et al., 2024), Chat-UniVi (13B) (Jin et al., 2024), Qwen2.5-VL (7B) (Wang et al., 2024), Qwen2.5-Omni (7B) (Xu et al., 2025), Qwen3-VL (8B) (Yang et al., 2025a) The model sizes range from 7B to 13B parameters for the language backbone, with the exception of VideoLLaMA2-72B (Cheng et al., 2024). All models are evaluated in a zero-shot setting. We utilize the vision and audio heads provided with the checkpoints to process the input. For InstructBLIP (Dai et al., 2023), an image model, we sample 4 frames from each video, process these frames with the image encoder and a Q-Former (Zhang et al., 2023c) as individual images, and then treat all features as a long sequence of image tokens for the language model.

Additionally, we evaluate a closed-source model, GPT-4o (Hurst et al., 2024) and Gemini-2.5-Flash (Comanici et al., 2025), in a zero-shot fashion. For GPT-4o, videos are preprocessed by uniformly selecting 4 frames from each video and down-scaling the resolution to half. The query used to prompt GPT-4o is provided in the appendix.

**Evaluation Metrics**: Since the answers in QIVD are in free-form, we determine the correctness of an answer using an LLM judge that receives a question, the ground-truth answer, and the predicted answer, alongside the short answer and the category of the question, and determines if the predicted answer is correct. We use a pre-trained Qwen3-8B model (Yang et al., 2025a) as the LLM judge (see Appendix C.6 for comparisons with other LLM judges). The prompts that were used are provided in the appendix. In addition, we report Bert (Zhang* et al., 2020), METEOR (Lavie & Agarwal, 2007), BLEU (Papineni et al., 2002), and ROUGE (Lin, 2004) scores between the ground-truth answers and the predicted answers.

## 5.2 RESULTS

We now present the results obtained from the three settings described in Section 5.1.

**Streaming setup**:    Table 4 presents the transcription results obtained from Whisper-Streaming (Macháček et al., 2023), where the transcription quality is quantified using BLEU (Papineni et al., 2002), ROUGE (Lin, 2004), and METEOR (Lavie & Agarwal, 2007) scores, by comparing the transcribed questions to the ground-truth questions. The when-to-answer metric, denoted by $\Delta t$ in the table, is measured as the Mean Absolute Error between the time-to-answer extracted by Whisper-Streaming and the ground-truth value. In addition to the overall MAE, we also report the

Table 5: Evaluation of baseline LMMs on the QIVD dataset using (a) questions and estimated when-to-answer timestamps by Whisper (Radford et al., 2023) and (b) ground-truth questions and timestamps. Corr. represents correctness by the LLM judge.

| Model | ASR Questions and Timestamps | | | | | Human Questions and Timestamps | | | | |
|---|---|---|---|---|---|---|---|---|---|---|
| | Corr. ↑ | BERT ↑ | METEOR ↑ | BLEU ↑ | ROUGE-L ↑ | Corr. ↑ | BERT ↑ | METEOR ↑ | BLEU ↑ | ROUGE-L ↑ |
| Chat-UniVi (Jin et al., 2024) | 34.66 | 89.94 | 37.47 | 6.08 | 28.45 | 40.79 | 90.50 | 40.02 | 7.24 | 31.22 |
| InstructBLIP (Dai et al., 2023) | 35.03 | 82.19 | 4.35 | 0.02 | 10.00 | 39.14 | 82.03 | 4.54 | 0.07 | 10.72 |
| LLaMA-VID (Li et al., 2024c) | 39.41 | 90.51 | 37.19 | 5.84 | 29.80 | 43.0 | 90.78 | 37.55 | 5.42 | 29.82 |
| LLaVA-NeXT (Liu et al., 2024a) | 19.45 | 85.29 | 22.85 | 1.38 | 11.64 | 22.66 | 85.78 | 24.50 | 1.67 | 13.22 |
| Video-ChatGPT (Maaz et al., 2024) | 32.45 | 90.53 | 38.13 | 7.58 | 31.08 | 36.59 | 91.01 | 40.59 | 9.07 | 33.58 |
| VideoChat (Li et al., 2024a) | 3.69 | 85.05 | 23.48 | 1.08 | 12.22 | 3.52 | 85.20 | 24.39 | 1.03 | 12.54 |
| VideoChat2 (Li et al., 2024b) | 44.66 | 91.13 | 45.49 | 11.35 | 41.38 | 50.35 | 91.52 | 47.93 | 12.43 | 43.87 |
| Video-LLaVA (Zhu et al., 2023; Lin et al., 2023) | 20.28 | 87.77 | 27.15 | 1.98 | 19.31 | 15.0 | 83.38 | 2.90 | 0.00 | 15.66 |
| VideoLLaMA (Zhang et al., 2023a) | 30.76 | 89.50 | 39.06 | 7.62 | 30.84 | 35.93 | 90.45 | 43.88 | 9.86 | 34.93 |
| VideoLLaMA2-7B (Cheng et al., 2024) | 43.34 | 91.18 | 47.20 | 13.93 | 40.63 | 50.07 | 91.71 | 51.08 | 16.41 | 43.97 |
| VideoLLaMA2-72B (Cheng et al., 2024) | 46.52 | 91.42 | 46.58 | 14.03 | 41.70 | 50.83 | 92.29 | 51.13 | 16.12 | 45.76 |
| VideoLLaMA3-7B (Zhang et al., 2025) | 50.59 | 90.92 | 45.20 | 11.21 | 40.54 | 56.38 | 91.63 | 48.56 | 12.72 | 43.84 |
| VideoLLM-online (Chen et al., 2024) | 17.97 | 76.60 | 27.36 | 2.81 | 20.39 | 23.62 | 88.45 | 33.08 | 3.99 | 25.26 |
| Flash-VStream (Zhang et al., 2024) | 44.28 | 89.85 | 28.95 | 4.17 | 27.05 | 49.59 | 90.48 | 30.79 | 5.05 | 29.90 |
| Qwen2.5-VL-7B (Wang et al., 2024) | 44.90 | 87.17 | 34.95 | 3.88 | 26.52 | 50.62 | 87.58 | 37.37 | 4.66 | 29.44 |
| Qwen2.5-Omni-7B (Xu et al., 2025) | 43.97 | 86.65 | 33.45 | 2.77 | 20.57 | 45.90 | 86.73 | 33.98 | 2.87 | 20.98 |
| Qwen3-VL-8B (Yang et al., 2025a) | 53.72 | 87.08 | 33.9 | 5.29 | 31.53 | 60.07 | 87.58 | 36.72 | 6.64 | 35.89 |
| Gemini-2.5-Flash (Comanici et al., 2025) | – | – | – | – | – | 58.07 | 90.43 | 43.07 | 8.33 | 36.05 |
| GPT-4o (Hurst et al., 2024) | – | – | – | – | – | 58.76 | 89.36 | 51.18 | 15.72 | 42.55 |
| Human (subset) | – | – | – | – | – | 87.33 | 93.01 | 53.21 | 17.40 | 49.76 |

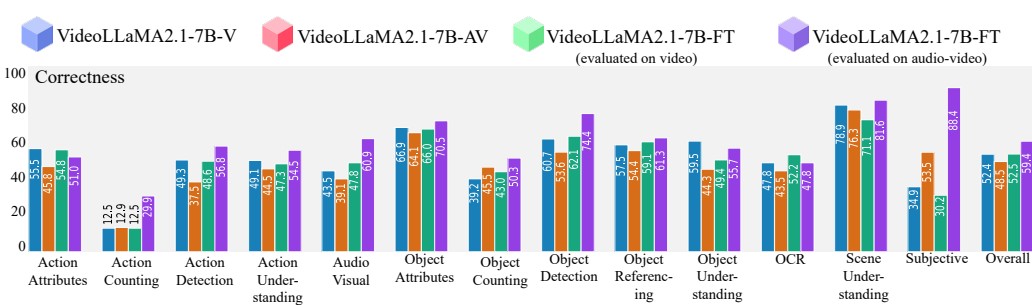

Figure 2: Evaluations of the public and finetuned VideoLLaMA2.1-7B-AV (Cheng et al., 2024) in vision + audio and vision-only settings.

mean values of both negative and positive $\Delta t$ instances. Notably, the average negative $\Delta t$ is larger in magnitude, suggesting that models which initiate responses immediately after detecting the end of a question tend to answer prematurely—often before sufficient contextual information has been received. We also report the results obtained from the standard Whisper model (Radford et al., 2023) as an additional baseline. It is worth noting that this model does not return any timestamps alongside the transcriptions.

The baseline LMMs are evaluated using a video trimmed at the when-to-answer timestamp and the question, both extracted via Whisper-Streaming (Machácek et al., 2023). Table 5 summarizes the baseline results.

**Offline setup**: For the evaluation in the offline setup, we provide the baseline LMMs with a video that is trimmed at the ground-truth when-to-answer timestamp alongside a ground-truth question. We summarize these results in Table 5. Additionally, we engage a non-expert human annotator to re-annotate a random subset of the dataset containing 300 samples, establishing a human baseline.

**Impact of audio**: The only publicly available checkpoint from the VideoLLaMA (Cheng et al., 2024) family that supports concurrent audio and video processing is VideoLLaMA2.1-7B-AV (Cheng et al., 2024). We evaluate this model using ground-truth transcribed questions in two distinct settings. In the first setting, we provide the model with both audio and visual information, while in the second setting, we supply only visual information. The results, depicted in Figure 2, show setting (1) in red and setting (2) in blue. Interestingly, and contrary to expectations, the model's performance degrades with the addition of audio information.

We fine-tune this model on QIVD using both audio and video modalities. Due to the dataset's small size, we apply 5-fold cross-validation. The vision encoder remains frozen, while the LLM backbone and audio pathway are fine-tuned for two epochs per fold. We repeat the initial experiments with

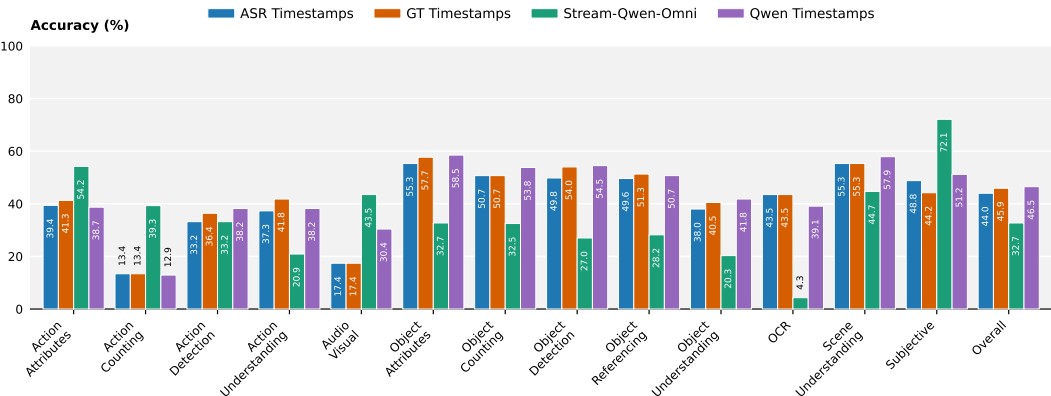

Figure 3: Evaluations of Qwen2.5-Omni (7B) with when-to-answer moment provided from different sources.

the fine-tuned model. Results in Figure 2 show setting (1) in purple and setting (2) in green . The fine-tuned model performs best when both modalities are available but underperforms the pretrained model in some video-only cases, likely due to its adaptation to multimodal inputs. Since QIVD relies heavily on audio cues, missing audio during inference significantly impairs performance.

**Impact of when-to-answer**: We adapted Qwen2.5-Omni (7B) (Xu et al., 2025) to support a streaming format by feeding the model chunked audio data, with each chunk representing one second. The model was fine-tuned to detect when-to-answer by emitting a special token during listening and generating a response once the appropriate moment was detected. Technical details are provided in Appendix D.3. The when-to-answer accuracy of Stream-Qwen-Omni is reported in Table 4, showing a significant improvement over Whisper-Streaming. To assess the impact of accurate timing, we evaluated Qwen2.5-Omni using timestamps from three sources: ground-truth (GT) annotations, ASR-Stream predictions, and Stream-Qwen2.5-Omni predictions. We also report the correctness of answers generated by Stream-Qwen-Omni. As shown in Figure 3, precise estimation of the when-to-answer moment substantially improves overall model performance.

## 5.3 DISCUSSIONS

To facilitate a more comprehensive analysis of the strengths and weaknesses of the baseline LMMs, we compare the correctness of selected baseline LMMs across individual categories of QIVD, as illustrated in Figure 4. The human baseline is derived from a small subset of the data, as detailed in Section 5.2. As demonstrated in Table 5 and Figure 4, there is a significant performance gap between a non-expert human and all the models, including state-of-the-art systems, across all evaluation categories. Humans demonstrate near-perfect performance in categories where AI systems struggle significantly, particularly in action counting, audio-visual integration, and object referencing. This disparity is most pronounced in tasks requiring temporal reasoning and deictic reference resolution, where humans outperform the best AI system by a large margin.

Furthermore, as shown in Figure 4, the baseline models exhibit inconsistent capabilities when faced with various types of situated visual reasoning. While these models perform reasonably well on basic object detection tasks, their performance declines markedly on tasks involving action counting, temporal sequencing, and audio-visual integration. This capability gap indicates that current models are optimized for static scene understanding rather than the dynamic temporal reasoning required for real-time interaction scenarios.

The most common failure modes include: (1) misinterpreting deictic references, (2) incorrect action counting, (3) temporal sequencing confusion, and (4) audio-visual misalignment. Many of these failures occur regardless of model size or architecture type, suggesting fundamental limitations in current approaches to multi-modal integration rather than just capacity constraints.

Our fine-tuning experiments show that the performance improvements from fine-tuning are not distributed uniformly across task categories. As shown in Figure 2, fine-tuning produces the most dramatic gains in action counting (+16.96%), action understanding (+10.00%), subjective (+23.26%), and audio-visual (+17.39%) tasks, while yielding minimal improvements in object attributes (+1.24%) and scene understanding (+2.63%). This asymmetric benefit pattern suggests that certain situated understanding capabilities are more amenable to data-driven adaptation than others. Particularly, even after fine-tuning, performance on action counting remains very low (29.91%), indicating that these temporal reasoning capabilities may require more sophisticated architectural inductive biases.

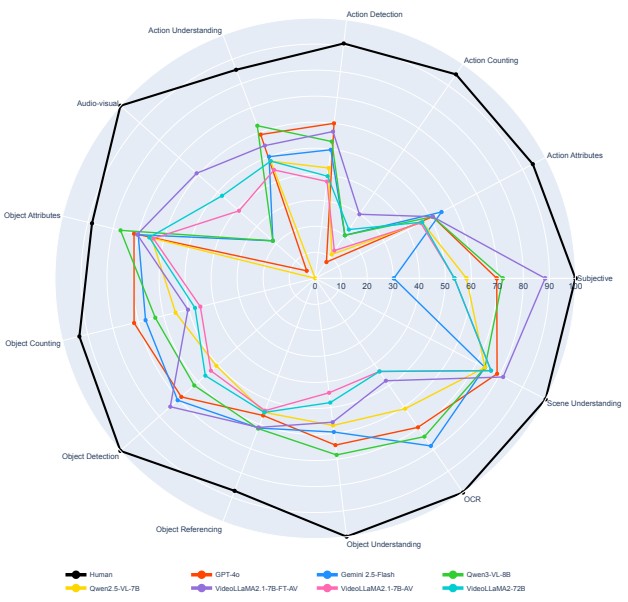

Figure 4: Correctness of selected baseline LMMs across QIVD categories (**best viewed with zoom**).

As shown in Figure 2, the integration of audio and visual modalities results in substantial performance gains across nearly all task categories. The VideoLLaMA2.1-7B-AV model shows a significant improvement over its vision-only counterpart in audio-visual tasks as we would expect. However, this improvement extends beyond explicitly audio-related tasks, with notable gains in subjective (+37.61%), object detection (+9.48%), and object counting (+10.14%). These findings empirically confirm our hypothesis that existing vision-language systems are fundamentally limited by their modular pipelines that process visual and audio information separately. We show that end-to-end multimodal training creates emergent capabilities that transcend simple feature concatenation, enabling more sophisticated situated understanding in real-time interactions.

## 6 CONCLUSION

We introduce QIVD, a comprehensive benchmark and dataset designed to assess and train LMMs (video, audio, and language) on a wide variety of tasks that require responding to humans in real time. Through extensive experiments, we identify key challenges with existing models for situated visual understanding. Our dataset follows a simple question-answering paradigm and thereby tests for situated understanding capabilities without being confounded by the need for multi-hop conversational capabilities. The dataset also does not require any domain-specific knowledge or complex reasoning skills. Yet we show that the task is still highly challenging for LMMs. Based on these insights, we hope that QIVD will inspire and guide future research, driving the development of AI systems that can interact with humans in realistic scenarios in an online fashion.

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

# Supplementary Material for QIVD

SUPPLEMENTARY CONTENTS

# A   SOCIAL IMPACTS OF QIVD

## A.1   LIMITATIONS OF QIVD

While our experiments with QIVD indicate that it presents a significant challenge for current multi-modal models—and despite the dataset being human-validated for annotation accuracy—there are several potential limitations and sources of bias to consider:

1. Relatively small class sizes, which may limit the diversity of questions and answers; 2. Recordings conducted in controlled environments, potentially reducing variability in lighting, background, and camera angles, which may affect model generalization; 3. Possible demographic biases in terms of gender, age, and ethnicity, which could impact model performance across diverse user groups.

## A.2   PRIVACY AND ETHICS IN QIVD

The data was collected under direct agreements with crowd workers, permitting both research and commercial use, ensuring compliance with applicable privacy regulations, including GDPR-equivalent standards. All videos were manually reviewed to identify and exclude any content containing issues such as the presence of individuals in the background. Personally identifiable information, including metadata, was removed to the extent possible to ensure participant privacy. Additionally, all contributors received fair and appropriate compensation in accordance with the standards of their respective regions. All contributors signed a consent form that explicitly permits both research and commercial use of their video and audio data, including use in training AI models. We will provide a contact email on the dataset release page, allowing participants to request data removal at any time.

## A.3   BROADER IMPACT OF QIVD

In addition to the aforementioned sources of bias, language models trained on QIVD may generate harmful or biased content, propagate misinformation, or offer inappropriate advice. These risks must be carefully considered when interacting with, deploying, or building upon such models.

# B   ADDITIONAL DATASET DETAILS

## B.1   ADDITIONAL EXAMPLES

We show additional video examples from our dataset in Figure B.1 to demonstrate the diversity of examples in QIVD.

## B.2   SEMANTIC TAXONOMY

A detailed definition of the categories used in QIVD is provided here.

**Action Attribute:** Inquiries regarding the manner in which an action was performed, such as *Which hand did I use to wave?* or *How fast did I jump?*—tests ability to recognize fine-grained characteristics of dynamic events.

**Action Counting:** Questions about the frequency of an action's repetition, such as *How many times did I clap?*—evaluates temporal reasoning and event segmentation capabilities.

**Action Detection:** Identifying the specific action that was performed, such as *What am I doing right now?*—assesses basic activity recognition in dynamic scenes.

**Action Understanding:** Questions about the purpose or outcome of an action, such as *What does this gesture mean?* or *Why am I moving the chair?*—tests higher-level action interpretation and intention recognition.

**Object Attributes:** Inquiries about the characteristics of an object, such as *What color is this book?* or *Is this cup empty or full?*—evaluates fine-grained visual perception of static properties.

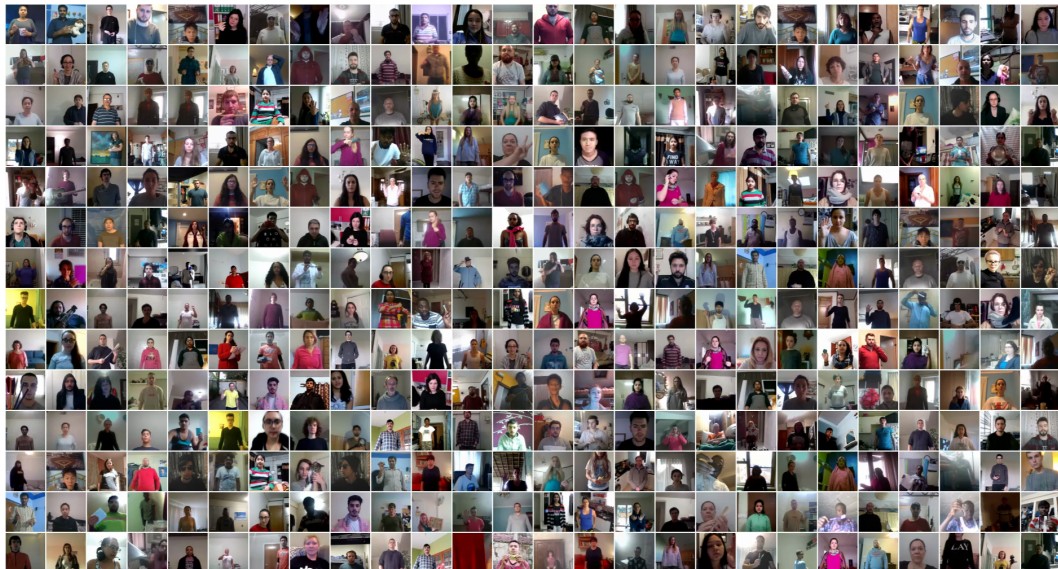

Figure B.1: Each image showcases a different video from our collection, demonstrating the substantial variation in visual scenarios captured within the dataset. These examples highlight the diversity of environments (indoor and outdoor settings), participants, objects, actions, lighting conditions, camera angles, and compositional elements present across the dataset.

**Object Counting:** Determining the number of objects present, such as *How many pens are on the table?*—tests quantitative reasoning and object individuation.

**Object Detection:** Identifying an object within the scene, such as *Is there a lamp in this room?*—assesses basic object recognition capabilities.

**Object Referencing:** Indirectly pointing to an object within the scene, such as *What am I pointing at?* or *What is behind me?*—evaluates spatial reasoning and deictic reference resolution.

**Object Understanding:** Questions about the nature or function of an object, such as *What is this tool used for?*—tests semantic knowledge about objects beyond mere recognition.

**Scene Understanding:** Inquiries about the environment, such as *What room am I in?* or *Is it daytime or nighttime?*—evaluates holistic scene interpretation.

**Audio-Visual:** Questions that require audio information for a complete answer, such as *What sound am I making?* or *Am I speaking loudly or softly?*—tests cross-modal integration capabilities.

**OCR:** Extracting text from an object, such as *What does this sign say?*—evaluates the capability to recognize text in the real world and within the context of the conversation.

**Subjective:** Soliciting general opinions about an object or scene, such as *Does this outfit look good?*—tests a model's ability to respond sensibly to subjective questions.

## B.3 WHEN-TO-ANSWER STATISTICS

Figure B.2 plots, for every clip, the temporal offset between the moment an answer becomes valid and the end of the video. Most questions are answerable within the last quarter of the clip, yet the long, asymmetric tail indicates that a non-trivial fraction require substantially earlier or later responses, confirming the need for models to reason over the full temporal span rather than assume a fixed "answer now" point.

We also quantify how often the correct answer becomes valid only after the question has finished. Because ground-truth end-of-question timestamps are unavailable, we use the end-of-question detected by Streaming-Whisper as a proxy. Table B.1 reports the count and fraction of clips whose

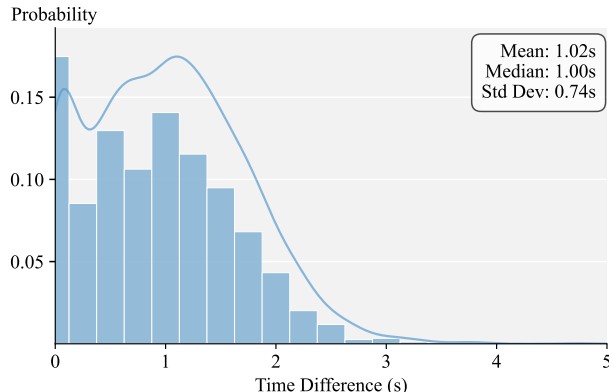

Figure B.2: Distribution of *optimal answer time* relative to the end of each clip. A value of $x$ on the horizontal axis means the ground-truth "answer now" moment occurs $x$ seconds before the video finishes.

Table B.1: Time difference statistics between ASR end-of-question and ground-truth when-to-answer.

| Time Threshold | Count | Percentage |
|---|---|---|
| $\geq 0.0$ s after | 2054 | 100.0% |
| $\geq 0.5$ s after | 1351 | 65.8% |
| $\geq 1.0$ s after | 745 | 36.3% |
| $\geq 1.5$ s after | 375 | 18.3% |
| $\geq 2.0$ s after | 186 | 9.1% |
| $\geq 2.5$ s after | 90 | 4.4% |
| $\geq 3.0$ s after | 34 | 1.7% |
| $\geq 4.0$ s after | 5 | 0.2% |

ground-truth when-to-answer time occurs at least a given threshold after the ASR end-of-question time.

Figure B.3 shows the empirical distribution of $\Delta t$, the error of our streaming-ASR estimate relative to the human "when-to-answer" annotation. The skew toward negative values reveals a systematic tendency of the ASR system to answer questions prematurely, often before sufficient visual context is available. Together, the figures highlight both the variability of optimal answer timing in real-world interactions and the practical challenge of detecting that moment reliably in a streaming setting.

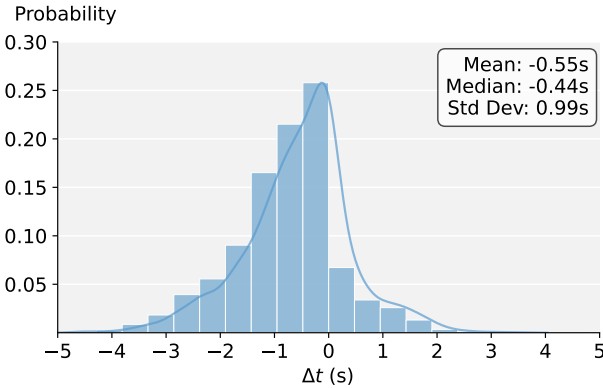

Figure B.3: Histogram of the signed error $\Delta t = t_{\mathrm{ASR}} - t_{\mathrm{GT}}$ between the streaming-ASR trigger and the human "answer now" annotation.

Table B.2: Facial expression and gesture recognition samples.

| Question | Answer |
|---|---|
| Do I look tired? | You seem tired. |
| What I'm doing right now? | You are smiling. |
| Is this monkey happy or sad? | The monkey on the pillow appears to be happy. |
| Am I happy? | It appears that you are happy. |
| What expression is this? | An expression of sadness or disappointment. |
| Did I approve or disapprove? | You gave a thumbs up, indicating your approval. |
| Am I smiling? | Yes, you are smiling. |
| Am I cold? | It seems that you are feeling cold. |
| Am I thinking? | It seems like you are deep in thought. |
| Did it hurt? | Yes based on your expression it did hurt. |
| Am I approving? | Yes, its seems like you are giving your approval with the thumbs up gesture. |
| Did I sleep enough last night? | You do not look sleep deprived. |
| Am I smiling? | No, you're not smiling. |
| What action is this? | You appear to be afraid of something. |
| What am I doing? | You are raising your arms showing a victory or happiness gesture. |

Table B.3: Evaluation of representative baseline LMMs on facial expression and gesture recognition samples of QIVD. Corr. represents the correctness score calculated by the LLM judge.

| Model | Corr. ↑ | BERT ↑ | METEOR ↑ | BLEU ↑ | ROUGE-L ↑ |
|---|---|---|---|---|---|
| Qwen2.5-Omni-7B (Xu et al., 2025) | 33.90 | 86.33 | 25.16 | 0.51 | 12.57 |
| Qwen2.5-VL-7B (Wang et al., 2024) | 44.07 | 86.78 | 29.35 | 0.83 | 17.23 |
| Qwen3-VL-8B (Yang et al., 2025a) | 61.02 | 86.81 | 28.49 | 0.73 | 19.7 |
| Gemini-2.5-Flash (Comanici et al., 2025) | 33.90 | 88.74 | 32.10 | 3.32 | 21.05 |
| GPT-4o (Hurst et al., 2024) | 32.20 | 85.48 | 36.98 | 2.44 | 21.28 |

## B.4 FACE-TO-FACE SAMPLES

Understanding facial expressions and gestures is a critical capability for any personal assistant system. To illustrate that QIVD effectively incorporates this functionality, we identified all samples within QIVD that are directly or indirectly associated with facial expression interpretation and gesture detection. In total, these comprise 59 samples, distributed as follows: 4 from action counting, 20 from action detection, 20 from action understanding, 10 from object attributes, 2 from object referencing, 2 from scene understanding, and 1 from subjective categories. A curated set of representative examples is presented in Table B.2. Furthermore, we evaluate the performance of representative models on these samples, using ground truth questions and timestamps. The corresponding results are summarized in Table B.3.

## C ADDITIONAL EXPERIMENTS

### C.1 COMPARING QIVD WITH OTHER DATASETS

We examine whether QIVD overlaps visually or semantically with prior video–QA corpora by embedding every clip with the *Cosmos-CV8×8×8* tokenizer (NVIDIA, 2025; NVIDIA et al., 2025) and measuring distances to clips from the closest public datasets: AVSD (DSTC7) (Alamri et al., 2018), and Social-IQ (Zadeh et al., 2019). For each video, we first normalize the frame-rate to 8 FPS and truncate (or zero-pad) to a maximum of 64 frames (8 seconds). The input frames are resized to $224 \times 224$ pixels. The resulting tensor, after batching and permuting to a $\mathcal{V} \in \mathbb{R}^{1 \times 3 \times 64 \times 224 \times 224}$ (Batch × Channels × Time × Height × Width) format, is passed through the tokenizer's encode method. This yields a continuous latent representation. We average these latent features across the temporal and spatial dimensions (dimensions 2, 3, and 4), obtaining a single embedding vector $\mathbf{e}$ per clip.

Table C.1 reports, for each clip in QIVD, the L2 distance to its nearest neighbour within QIVD itself as well as to AVSD (Alamri et al., 2018) and Social-IQ (Zadeh et al., 2019). The intra-QIVD distances

Table C.1: Nearest-neighbor L2 distances between QIVD clips and each dataset.

| Comparison | Mean | Median | Min | Max | 5th Percentile |
|---|---|---|---|---|---|
| QIVD vs. QIVD | 0.0157 | 0.0148 | 0.0031 | 0.1124 | 0.0062 |
| QIVD vs. AVSD | 0.0386 | 0.0369 | 0.0125 | 0.1238 | 0.0173 |
| QIVD vs. Social-IQ | 0.0894 | 0.0871 | 0.0257 | 0.2156 | 0.0458 |

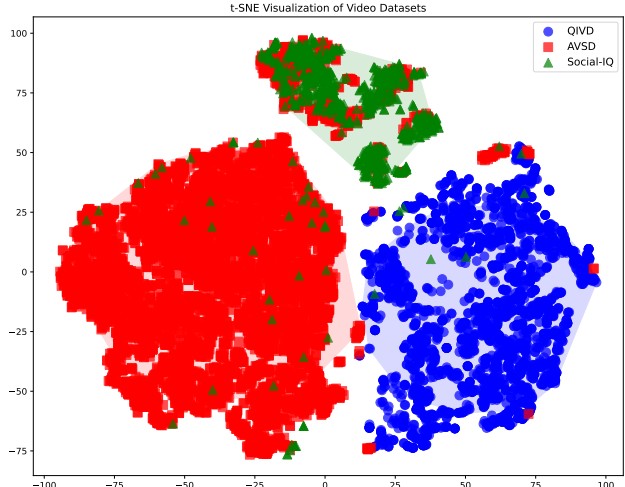

Figure C.1: Two-dimensional t-SNE projection of the 1024-dimensional embeddings for QIVD (blue), AVSD (red), and Social-IQ (green). QIVD clips form a tight, coherent cluster that is clearly separated from those of AVSD and Social-IQ, illustrating their distinct distributions in latent space.

(mean 0.0157, 5th percentile 0.0062) are smaller than the inter-dataset distances: AVSD (mean 0.0386, 5th percentile 0.0173) and Social-IQ (mean 0.0894, 5th percentile 0.0458). Only a small fraction of QIVD clips find their closest counterpart outside the test split indicating minimal overlap with prior benchmarks and underscoring that QIVD brings substantially novel visual–semantic content.

Figure C.1 further illustrates this separation in a two-dimensional t-SNE projection of the 1024-dimensional embeddings: QIVD points form a tight cluster on the right, clearly distinct from AVSD (red) and Social-IQ (green), which occupy disjoint regions. We demonstrate that QIVD is significantly different from prior datasets.

## C.2    CORRELATION WITH STANDARD METRICS

In Table C.2, we report the correlation between standard metrics (BLEU, ROUGE-L, METEOR, BERTScore) and the LLM Judge correctness score. We observe that ROUGE-L ($r = 0.774$) and METEOR ($r = 0.678$) correlate relatively well with the judge, while BERTScore ($r = 0.616$) shows a lower correlation.

Table C.2: Correlation between standard metrics (BLEU, ROUGE-L, METEOR, BERTScore) and the LLM Judge Correctness score across all evaluated models.

| Metric | Pearson ($r$) | Spearman ($\rho$) |
|---|---|---|
| BLEU-4 | 0.675 | 0.680 |
| ROUGE-L | 0.774 | 0.737 |
| METEOR | 0.678 | 0.688 |
| BERTScore | 0.616 | 0.506 |

Table C.3: Correctness (%) comparison between Static (Object-related) and Temporal (Action/Scene-related) categories. Humans maintain consistency, whereas models show a sharp decline in temporal tasks.

| Model | Static Categories | Temporal Categories | Gap |
|---|---|---|---|
| Human | 88.44 | 86.46 | **-1.98** |
| GPT-4o | 65.24 | 46.06 | -19.18 |
| Gemini-2.5-Flash | 66.10 | 44.14 | -21.96 |
| Qwen3-VL-8B | 67.06 | 46.36 | -20.70 |
| Qwen2.5-VL-7B | 57.69 | 36.97 | -20.72 |
| VideoLLaMA3-7B | 62.40 | 45.25 | -17.15 |

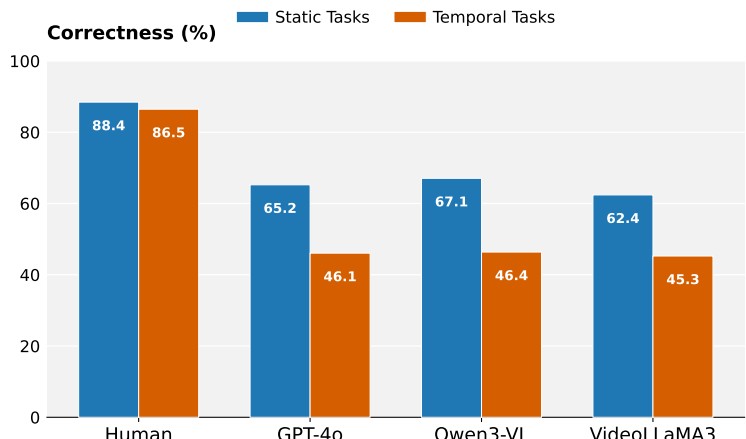

Figure C.2: Comparison of model performance on Static vs. Temporal tasks. While humans perform equally well on both, all models show a significant performance degradation on temporal tasks.

## C.3    STATIC VS. TEMPORAL REASONING GAP

We analyze the performance gap between "Static" categories (Object Attributes, Counting, Detection, Referencing, Understanding, OCR) and "Temporal" categories (Action Attributes, Counting, Detection, Understanding, Audio-Visual, Scene Understanding). Table C.3 reveals a striking disparity: while human performance remains consistent across both types ($\sim$87-88%), all state-of-the-art models suffer a significant performance drop ($> 19\%$) when moving from static to temporal reasoning tasks. We observe that current LMMs are biased towards static image capabilities and struggle with the dynamic nature of real-world video interactions.

## C.4    CATEGORY-WISE PERFORMANCE

Table C.4 provides the granular correctness scores for each semantic category across the top-performing models. We observe that even the best models achieve $< 35\%$ on some categories, such as Action Counting, compared to 85.7% for humans.

## C.5    USING DIFFERENT VIDEO SAMPLING STRATEGIES

We study how restricting visual evidence to a short temporal window around the moment the question is spoken affects performance. For a given clip, we consider the segment spanning $\pm 0.5$, $\pm 1.0$, $\pm 2.0$, $\pm 3.0$, $\pm 4.0$, or $\pm 5.0$ seconds around the question timestamp and uniformly sample frames from that segment. We evaluate Qwen2.5-VL-7B under these settings as well as the full-video baseline. Results are reported in Table C.5. Consistent with the intuition that many questions require context before and after the utterance, very short windows harm performance. Wider windows recover accuracy, with $\pm 3$–$\pm 4$ seconds yielding the best results; beyond $\pm 5$ seconds, inputs effectively cover the full clip and performance plateaus.

Table C.4: Detailed breakdown of Correctness (%) by category for top-performing models and Human baseline (Offline/GT setting).

| Category | Human | GPT-4o | Gemini-2.5 | Qwen3-VL | Qwen2.5-VL | VideoLLaMA3 |
|---|---|---|---|---|---|---|
| Action Attributes | 83.33 | 50.97 | 54.84 | 50.97 | 46.45 | 47.10 |
| Action Counting | 85.71 | 7.59 | 20.09 | 20.09 | 11.16 | 33.48 |
| Action Detection | 88.64 | 60.00 | 49.77 | 52.95 | 42.73 | 48.18 |
| Action Understanding | 85.71 | 59.09 | 50.00 | 62.73 | 48.18 | 47.27 |
| Object Attributes | 78.33 | 71.71 | 70.05 | 77.05 | 64.77 | 71.35 |
| Object Counting | 96.67 | 71.68 | 67.13 | 63.29 | 55.24 | 50.70 |
| Object Detection | 95.24 | 68.72 | 70.62 | 62.09 | 50.71 | 66.35 |
| Object Referencing | 89.87 | 56.37 | 61.53 | 61.76 | 55.10 | 60.20 |
| Object Understanding | 100.00 | 64.56 | 59.49 | 68.35 | 56.96 | 54.43 |
| Scene Understanding | 75.00 | 78.95 | 73.68 | 73.68 | 73.68 | 73.68 |
| Audio-Visual | 100.00 | 4.35 | 21.74 | 21.74 | 0.00 | 34.78 |
| OCR | 100.00 | 69.57 | 78.26 | 73.91 | 60.87 | 47.83 |
| Subjective | 60.00 | 69.77 | 30.23 | 72.09 | 58.14 | 51.16 |

Table C.5: Effect of sampling frames within temporal windows around the question timestamp for Qwen2.5-VL-7B (Wang et al., 2024). Values are proportions or similarity scores (higher is better).

| Model | Corr. ↑ | BERT ↑ | METEOR ↑ | BLEU ↑ | ROUGE-L ↑ |
|---|---|---|---|---|---|
| Qwen2.5-VL-7B (full video) | 60.00 | 87.58 | 37.37 | 4.66 | 29.44 |
| Qwen2.5-VL-7B (sampled around question) $\pm 0.5$ s | 38.28 | 78.90 | 25.40 | 2.60 | 20.30 |
| Qwen2.5-VL-7B (sampled around question) $\pm 1.0$ s | 42.00 | 80.10 | 28.90 | 3.10 | 22.80 |
| Qwen2.5-VL-7B (sampled around question) $\pm 2.0$ s | 60.41 | 87.00 | 42.02 | 4.60 | 29.42 |
| Qwen2.5-VL-7B (sampled around question) $\pm 3.0$ s | 61.20 | 87.90 | 38.10 | 4.72 | 29.80 |
| Qwen2.5-VL-7B (sampled around question) $\pm 4.0$ s | 60.75 | 88.01 | 37.80 | 4.68 | 29.55 |
| Qwen2.5-VL-7B (sampled around question) $\pm 5.0$ s | 59.95 | 87.72 | 37.55 | 4.64 | 29.48 |

## C.6 LLM JUDGE ACCURACY

To evaluate the accuracy of the LLM judge, we randomly select a subset of 300 samples from QIVD and collect human ratings of GPT-4o's (Hurst et al., 2024) answers. In addition to the human evaluation, we use three automatic judges: LLaMA3-8B (et. al., 2024) and two recent Qwen3 (Yang et al., 2025a) models (32B and 8B). The fraction of answers deemed correct by each evaluator is reported in Table C.6. Based on Table C.6, we use Qwen3-8B (Yang et al., 2025a) as the main judge throughout our experiments. We provide the results with LLaMA3-8B (et. al., 2024) as the judge in Table C.7.

Table C.6: Correctness of GPT-4o answers on a 300-sample subset under different evaluators. Values are the proportion marked correct.

| Evaluator | Correctness |
|---|---|
| Human Evaluation | 0.64 |
| LLaMA3-8B (et. al., 2024) | 0.68 |
| Qwen3-32B (Yang et al., 2025a) | 0.57 |
| Qwen3-8B (Yang et al., 2025a) | 0.59 |

## C.7 FAILURE CASES

To further demonstrate the limitations of current LMMs in addressing routine real-life questions, we present a series of simple queries that, while effortlessly answered by human annotators, pose significant challenges for LMMs (see Figure C.3). Notably, these examples highlight the shortcomings of several advanced models, including the robust GPT-4o, the large-scale VideoLLaMA2-72B (Zhang et al., 2023a), and even the fine-tuned VideoLLaMA2.1-7B-AV (Zhang et al., 2023a).

Table C.7: Evaluation of baseline LMMs on the QIVD dataset using (a) questions and estimated when-to-answer timestamps by Whisper (Radford et al., 2023) and (b) ground-truth questions and timestamps. Corr. represents correctness by LLM judge with LLaMA3-8B (et. al., 2024) as the judge.

| Model | ASR Questions and Timestamps | | | | | Human Questions and Timestamps | | | | |
|---|---|---|---|---|---|---|---|---|---|---|
| | Corr. ↑ | BERT ↑ | METEOR ↑ | BLEU ↑ | ROUGE-L ↑ | Corr. ↑ | BERT ↑ | METEOR ↑ | BLEU ↑ | ROUGE-L ↑ |
| Chat-UniVi (Jin et al., 2024) | 39.69 | 89.94 | 37.47 | 6.08 | 28.45 | 45.10 | 90.50 | 40.02 | 7.24 | 31.22 |
| InstructBLIP (Dai et al., 2023) | 37.17 | 82.19 | 4.35 | 0.02 | 9.99 | 41.14 | 82.03 | 4.54 | 0.07 | 10.72 |
| LLaMA-VID (Li et al., 2024c) | 43.48 | 90.51 | 37.18 | 5.84 | 29.80 | 48.48 | 90.78 | 37.55 | 5.42 | 29.82 |
| LLaVA-NeXT (Liu et al., 2024a) | 24.97 | 85.29 | 22.85 | 1.38 | 11.64 | 28.90 | 85.78 | 24.50 | 1.67 | 13.22 |
| Video-ChatGPT (Maaz et al., 2024) | 35.38 | 90.53 | 38.14 | 7.58 | 31.09 | 40.76 | 91.01 | 40.59 | 9.07 | 33.58 |
| VideoChat (Li et al., 2024a) | 8.00 | 85.05 | 23.48 | 1.08 | 12.22 | 8.31 | 85.20 | 24.39 | 1.03 | 12.54 |
| VideoChat2 (Li et al., 2024b) | 46.07 | 91.13 | 45.49 | 11.35 | 41.38 | 53.07 | 91.52 | 47.93 | 12.43 | 43.87 |
| Video-LLaVA (Zhu et al., 2023; Lin et al., 2023) | 23.52 | 87.77 | 27.15 | 1.98 | 19.31 | 18.62 | 83.38 | 2.90 | 0.00 | 15.66 |
| VideoLLaMA (Zhang et al., 2023a) | 33.52 | 89.50 | 39.05 | 7.62 | 30.84 | 39.21 | 90.45 | 43.88 | 9.86 | 34.93 |
| VideoLLaMA2-7B (Cheng et al., 2024) | 44.31 | 91.18 | 47.20 | 13.93 | 40.63 | 52.69 | 91.71 | 51.08 | 16.41 | 43.97 |
| VideoLLaMA2-72B (Cheng et al., 2024) | 47.69 | 91.42 | 46.60 | 14.04 | 41.71 | 53.41 | 92.29 | 51.13 | 16.12 | 45.76 |
| VideoLLaMA3-7B (Zhang et al., 2025) | 52.31 | 90.92 | 45.20 | 11.21 | 40.55 | 59.62 | 91.63 | 48.56 | 12.72 | 43.84 |
| VideoLLM-online (Chen et al., 2024) | – | 76.60 | 27.36 | 2.81 | 20.39 | – | 88.45 | 33.08 | 3.99 | 25.26 |
| Flash-VStream (Zhang et al., 2024) | – | 89.85 | 28.95 | 4.17 | 27.05 | – | 90.48 | 31.49 | 5.05 | 29.90 |
| Qwen2.5-VL-7B (Wang et al., 2024) | 53.55 | 87.17 | 34.95 | 3.89 | 26.52 | 60.00 | 87.58 | 37.37 | 4.66 | 29.44 |
| Qwen2.5-Omni-7B (Xu et al., 2025) | 44.76 | 86.65 | 33.45 | 2.77 | 20.57 | 46.97 | 86.73 | 33.98 | 2.87 | 20.98 |
| Qwen3-VL-8B (Yang et al., 2025a) | – | 87.08 | 33.90 | 5.29 | 31.53 | 64.69 | 87.58 | 36.72 | 6.64 | 35.89 |
| Gemini-2.5-Flash (Comanici et al., 2025) | – | – | – | – | – | 66.41 | 90.43 | 43.07 | 8.33 | 36.05 |
| GPT-4o (Hurst et al., 2024) | – | – | – | – | – | 66.38 | 89.36 | 51.18 | 15.72 | 42.55 |
| Human (subset) | – | – | – | – | – | 89.33 | 93.01 | 53.21 | 17.40 | 49.76 |

Table C.8: ASR performance comparison.

| Model | METEOR ↑ | BLEU ↑ | ROUGE-L ↑ | $\Delta t \downarrow$ |
|---|---|---|---|---|
| Whisper (Radford et al., 2023) | $90.01 \pm 23.11$ | $80.95 \pm 35.13$ | $90.32 \pm 22.66$ | - |
| Whisper-Streaming (Machácek et al., 2023) | $92.34 \pm 15.31$ | $74.57 \pm 33.52$ | $91.82 \pm 15.72$ | $0.83 \pm 0.77$ |

## C.8 STATISTICAL SIGNIFICANCE

We report the standard deviation values corresponding to table 4 and table 5 in table C.8, table C.9, and table C.10.

Table C.9: Evaluation of baseline LMMs on the QIVD dataset using questions and when-to-answer timestamps extracted by Whisper-Streaming (Radford et al., 2023). Corr. represents the correctness score calculated by the LLM judge.

| Model | Corr. ↑ | BERT ↑ | METEOR ↑ | BLEU ↑ | ROUGE-L ↑ |
|---|---|---|---|---|---|
| Chat-UniVi (Jin et al., 2024) | 34.66 (±47.60) | 89.94 (±3.56) | 37.47 (±23.53) | 6.08 (±16.44) | 28.45 (±22.41) |
| InstructBLIP (Dai et al., 2023) | 35.03 (±47.72) | 82.19 (±3.00) | 4.35 (±6.53) | 0.02 (±0.73) | 9.99 (±14.40) |
| LLaMA-VID (Li et al., 2024c) | 39.41 (±48.87) | 90.51 (±3.56) | 37.18 (±23.25) | 5.84 (±16.39) | 29.80 (±22.03) |
| LLaVA-NeXT (Liu et al., 2024a) | 19.45 (±39.59) | 85.29 (±3.24) | 22.85 (±15.72) | 1.38 (±8.68) | 11.64 (±15.21) |
| Video-ChatGPT (Maaz et al., 2024) | 32.45 (±46.83) | 90.53 (±3.78) | 38.14 (±24.78) | 7.58 (±19.46) | 31.09 (±24.45) |
| VideoChat (Li et al., 2024a) | 3.69 (±18.85) | 85.05 (±2.77) | 23.48 (±15.29) | 1.08 (±6.47) | 12.22 (±12.29) |
| VideoChat2 (Li et al., 2024b) | 44.66 (±49.72) | 91.13 (±3.88) | 45.49 (±26.63) | 11.35 (±23.38) | 41.38 (±26.04) |
| Video-LLaVA (Zhu et al., 2023; Lin et al., 2023) | 20.28 (±40.21) | 87.77 (±3.37) | 27.15 (±18.88) | 1.98 (±9.73) | 19.31 (±17.63) |
| VideoLLaMA (Zhang et al., 2023a) | 30.76 (±46.16) | 89.50 (±4.56) | 39.05 (±26.06) | 7.62 (±18.87) | 30.84 (±24.83) |
| VideoLLaMA2-7B (Cheng et al., 2024) | 43.34 (±49.56) | 91.18 (±4.18) | 47.20 (±27.92) | 13.93 (±26.57) | 40.63 (±27.22) |
| VideoLLaMA2-72B (Cheng et al., 2024) | 46.52 (±49.89) | 91.42 (±5.68) | 46.60 (±28.88) | 14.04 (±27.41) | 41.71 (±28.50) |
| VideoLLaMA3-7B (Zhang et al., 2025) | 50.59 (±50.01) | 90.92 (±5.34) | 45.20 (±27.14) | 11.21 (±23.54) | 40.55 (±26.55) |
| VideoLLM-online (Chen et al., 2024) | 17.97 (±38.40) | 76.60 (±29.79) | 27.36 (±22.11) | 2.81 (±10.28) | 20.39 (±19.30) |
| Flash-VStream (Zhang et al., 2024) | 44.28 (±49.68) | 89.85 (±3.73) | 28.95 (±24.21) | 4.17 (±15.38) | 27.05 (±24.56) |
| Qwen2.5-VL-7B (Wang et al., 2024) | 44.90 (±49.75) | 87.17 (±2.71) | 34.95 (±20.21) | 3.89 (±10.62) | 26.52 (±23.25) |
| Qwen2.5-Omni-7B (Xu et al., 2025) | 43.97 (±49.64) | 86.65 (±1.95) | 33.45 (±17.12) | 2.77 (±5.94) | 20.57 (±12.71) |
| Qwen3-VL-8B (Yang et al., 2025a) | 53.72 (±49.87) | 87.08 (±3.08) | 33.90 (±22.11) | 5.29 (±12.70) | 31.53 (±27.10) |

## D ADDITIONAL EXPERIMENTAL DETAILS

### D.1 IMPLEMENTATION DETAILS

All experiments were conducted in PyTorch. Every open-source LMM checkpoint was loaded in half-precision (FP16), except for the 72B parameter VideoLLaMA2 (Zhang et al., 2023a) model,

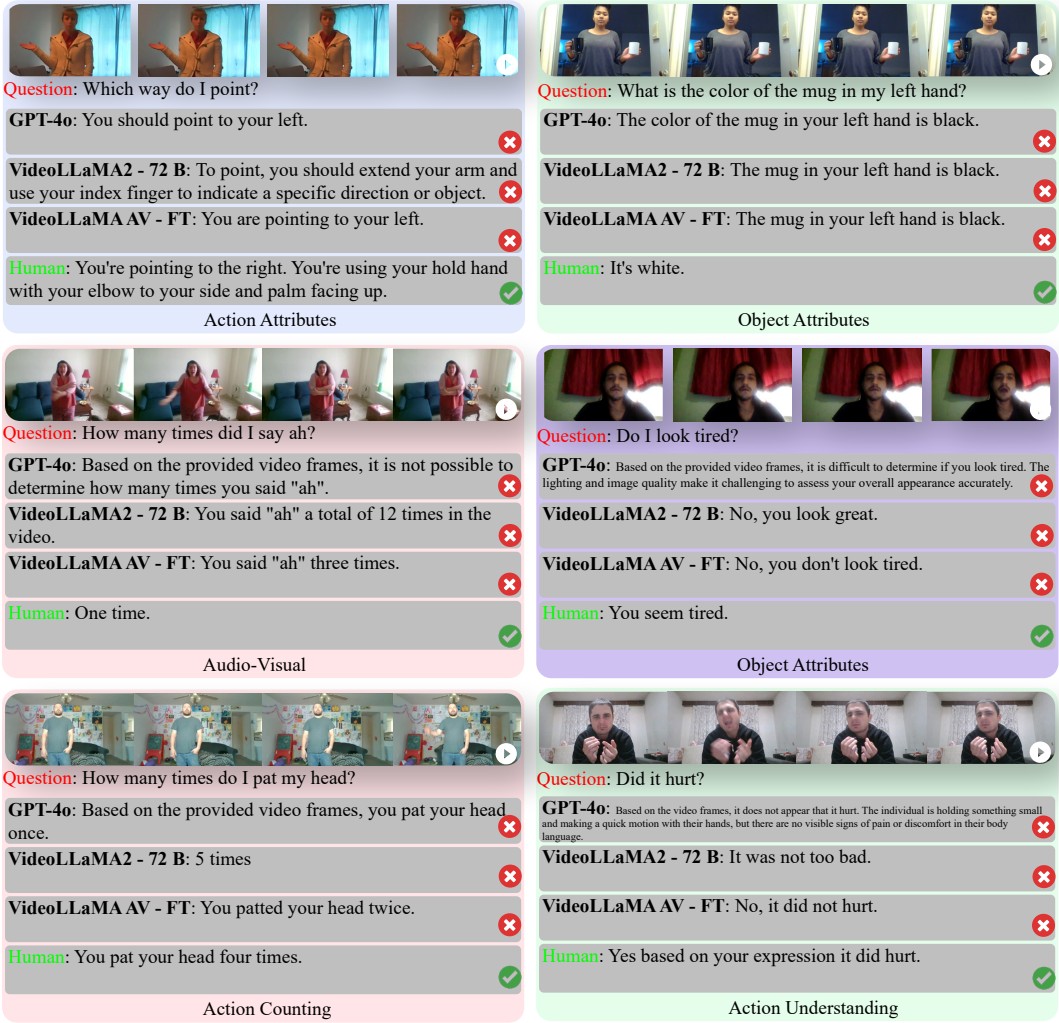

Figure C.3: Simple daily face-to-face questions that strong baseline LMMs such as GPT-4o, VideoLLaMA2-72B, and VideoLLaMA2.1-7B-AV fail to answer.

which was run with post-training INT8 quantization to satisfy memory limits. The inference code for each baseline was taken unmodified from the authors' public repositories and executed with the best-performing hyper-parameter settings provided by the authors. All of our experiments were run on a single A100-80 GB GPU.

## D.2    VIDEOLLAMA FINETUNING DETAILS

We initialize from the publicly released VideoLLaMA 2.1-7B-AV (Zhang et al., 2023a) checkpoint and reuse the authors' training recipe with minimal modifications. The 2900 clips in QIVD are partitioned into 5 non-overlapping folds via a deterministic hash of the video filename.Each fold in turn serves as validation, while the remaining four constitute the training split (~2.32K clips). The architecture components updated during fine-tuning are summarized in Table D.1. We present all the hyperparameters in Table D.2.

## D.3    STREAM-QWEN-OMNI DETAILS

This section provides details on how we convert the Qwen2.5-Omni model to a streaming format. The conversion is achieved by changing the way multi-modal data is provided to the model. We split the audio-visual data into 1-second chunks and feed the model one chunk at a time. The model is

Table C.10: Evaluation of baseline LMMs on the QIVD dataset using ground-truth questions and timestamps. Corr. represents correctness by LLM judge.

| Model | Corr. ↑ | BERT ↑ | METEOR ↑ | BLEU ↑ | ROUGE-L ↑ |
|---|---|---|---|---|---|
| Chat-UniVi (Jin et al., 2024) | 40.79 (±49.15) | 90.50 (±3.49) | 40.02 (±23.64) | 7.24 (±18.29) | 31.22 (±22.70) |
| InstructBLIP (Dai et al., 2023) | 39.14 (±48.81) | 82.03 (±3.13) | 4.54 (±6.81) | 0.07 (±1.70) | 10.72 (±14.56) |
| LLaMA-VID (Li et al., 2024c) | 43.00 (±49.52) | 90.78 (±3.32) | 37.55 (±22.42) | 5.42 (±15.59) | 29.82 (±21.12) |
| LLaVA-NeXT (Liu et al., 2024a) | 22.66 (±41.87) | 85.78 (±3.40) | 24.50 (±16.66) | 1.67 (±9.53) | 13.22 (±16.54) |
| Video-ChatGPT (Maaz et al., 2024) | 36.59 (±48.18) | 91.01 (±3.78) | 40.59 (±25.20) | 9.07 (±21.51) | 33.58 (±25.11) |
| VideoChat (Li et al., 2024a) | 3.52 (±18.42) | 85.20 (±2.72) | 24.39 (±15.51) | 1.03 (±5.52) | 12.54 (±12.11) |
| VideoChat2 (Li et al., 2024b) | 50.34 (±50.01) | 91.52 (±3.81) | 47.93 (±26.62) | 12.43 (±24.04) | 43.87 (±25.97) |
| Video-LLaVA (Zhu et al., 2023; Lin et al., 2023) | 15.00 (±35.71) | 83.38 (±1.85) | 2.90 (±5.27) | 0.00 (±0.00) | 15.66 (±16.00) |
| VideoLLaMA (Zhang et al., 2023a) | 35.93 (±47.99) | 90.45 (±4.15) | 43.88 (±25.81) | 9.86 (±21.99) | 34.93 (±25.09) |
| VideoLLaMA2-7B (Cheng et al., 2024) | 50.07 (±50.01) | 91.71 (±4.15) | 51.08 (±27.91) | 16.41 (±28.98) | 43.97 (±27.56) |
| VideoLLaMA2-72B (Cheng et al., 2024) | 50.83 (±50.00) | 92.29 (±4.35) | 51.13 (±27.95) | 16.12 (±28.86) | 45.76 (±28.06) |
| VideoLLaMA3-7B (Zhang et al., 2025) | 56.38 (±49.60) | 91.63 (±4.24) | 48.56 (±26.81) | 12.72 (±24.92) | 43.84 (±26.11) |
| VideoLLM-online (Chen et al., 2024) | 23.62 (±42.48) | 88.45 (±3.55) | 33.08 (±21.42) | 3.99 (±12.35) | 25.26 (±19.97) |
| Flash-VStream (Zhang et al., 2024) | 49.59 (±50.01) | 90.48 (±3.57) | 31.49 (±24.88) | 5.05 (±17.12) | 29.90 (±24.98) |
| Qwen2.5-VL-7B (Wang et al., 2024) | 50.62 (±50.00) | 87.58 (±2.63) | 37.37 (±20.46) | 4.66 (±11.67) | 29.44 (±24.18) |
| Qwen2.5-Omni-7B (Xu et al., 2025) | 45.90 (±49.84) | 86.73 (±1.93) | 33.98 (±17.22) | 2.87 (±5.96) | 20.98 (±12.71) |
| Qwen3-VL-8B (Yang et al., 2025a) | 60.07 (±48.98) | 87.58 (±3.00) | 36.72 (±22.77) | 6.64 (±14.11) | 35.89 (±28.07) |
| Gemini-2.5-Flash (Comanici et al., 2025) | 58.07 (±49.35) | 90.43 (±4.12) | 43.07 (±25.20) | 8.33 (±20.68) | 36.05 (±26.01) |
| GPT-4o (Hurst et al., 2024) | 58.76 (±49.24) | 89.36 (±15.25) | 51.18 (±27.32) | 15.72 (±28.27) | 42.55 (±28.17) |
| Human (subset) | 87.33 (±33.32) | 93.01 (±3.89) | 53.21 (±25.22) | 17.40 (±30.90) | 49.76 (±25.18) |

Table D.1: Trainable versus frozen modules during fine-tuning.

| Module | Status |
|---|---|
| Vision encoder (SigLIP-SO400M/16F (Zhai et al., 2023)) | ❄ Frozen |
| Audio tower (BEATs iter-3 (Chen et al., 2022) + AS-2M) | 🔥 Trainable |
| Multimodal projector (A) | 🔥 Trainable |
| Multimodal projector (V) | ❄ Frozen |
| LLM backbone (Qwen2-7B-Instruct (Yang et al., 2024)) | 🔥 Trainable |

fine-tuned to generate a special token (`"..."`) when it is listening and watching, and to produce an answer when it reaches the time-to-answer.

For this purpose, we reformat the training data as shown in the following example.

Original format:

```
{
    "time_to_answer": "3.8s",
    "answer": "You are holding a Rubik's cube in your left hand."
}
```

Streaming format:

```
[
    [0.0s, 1.0s, "..."],
    [1.0s, 2.0s, "..."],
    [2.0s, 3.0s, "..."],
    [3.0s, 4.0s, "You are holding a Rubik's cube in your left hand."],
    [4.0s, 5.0s, "..."],
    [5.0s, 6.0s, "..."]
]
```

A schematic of this architectural modification is shown in Figure D.1. This approach enables the Stream-Qwen-Omni model to detect when-to-answer with a granularity of one second.

During finetuning, all weights are kept frozen except for the vision adapter, audio adapter, and embedding layer. The network is trained with a batch size of 1, gradient accumulation steps of 1, and 2 full-size frames per second for 1 epoch. Other training details follow Table D.2. Similar to VideoLLaMA finetuning, we use 5-fold cross validation with the same splits.

Table D.2: Hyper-parameters and optimization settings for each cross-validation fold.

| Hyper-parameter | Value |
| --- | --- |
| Training precision | bf16 |
| Global batch size | 8 videos ($1 \times 8$) |
| Frames per clip | 8 |
| Epochs | 2 |
| Optimizer | AdamW (Loshchilov & Hutter, 2019) |
| Adam $(\beta_1, \beta_2, \varepsilon)$ | $(0.9, 0.999, 10^{-8})$ |
| Weight decay | 0 |
| Learning rate schedule | $2 \times 10^{-5} \rightarrow 0$ (cosine), 3% warm-up |
| Gradient accumulation steps | 8 |
| Gradient clip-norm | 1.0 |
| Distributed strategy | Deepspeed ZeRO-2 (parameter off-load) (Rajbhandari et al., 2020) |

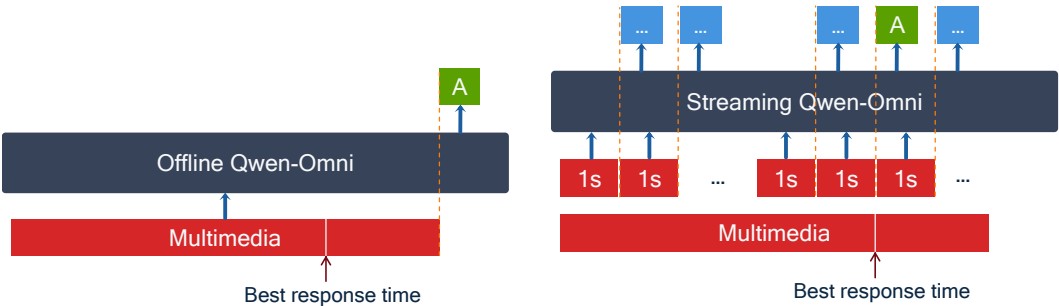

Figure D.1: Details of Stream-Qwen-Omni structure.

## D.4 LMM EVALUATION

The prompts supplied to the LLM-based judge are reproduced in Table D.3 and Table D.4. Since *subjective* questions require a qualitatively different notion of correctness, we evaluate those cases with a dedicated prompt that deems responses acceptable provided they are contextually appropriate, friendly, and affirmatively phrased.

## D.5 GPT-4O PROMPT

To process QIVD videos with GPT-4o, we uniformly select four frames from each video and spatially downscale them to half their original size. The preprocessed frames are then combined with the question into a query, as illustrated in Table D.5, and this query is used to prompt GPT-4o.

## D.6 GPT-4O REFUSAL CASES

: GPT-4o declines to answer 76 questions in QIVD due to `ResponsibleAIPolicyViolation`. Given that the samples in QIVD undergo extensive quality checks, the likelihood of samples violating the `ResponsibleAIPolicy` is very low. In these instances, GPT-4o mistakenly classifies the samples as `ResponsibleAIPolicyViolation` and refuses to provide an answer. We consider these cases, where GPT-4o provides an empty response, as incorrect in our evaluations. Examples of questions that GPT-4o refused to answer are shown in Figure D.2.

**General Correctness Evaluation**

**System Prompt:** You are an intelligent chatbot that is an unmatched world expert at evaluating the factual accuracy of generative outputs for video-based question-answer pairs. You are tasked with evaluating the correctness of a predicted answer by comparing it to a reference answer. The answers are to the same question. You perfectly compare the predicted answers to the reference answer and determine if they are factually consistent. As needed, you expertly consider the short version of the reference answer which contains only relevant details, and the question category.
You are a perfectionist at adhering to these criteria for correctness: Follow these steps:

- You are given the Question, the Category, the Reference Answer (short), the Reference Answer, and the Predicted Answer.
- Read the Question: Carefully read and understand the question provided.
- Read the Category: Take note of the category of the question to understand the context.
- Read the Reference Answer (short): Carefully read and understand the reference short answer that contains the key point.
    - If the short answer is 'NA', IGNORE the short answer.
- Read the Reference Answer: Carefully read and understand the reference answer provided.
- Read the Predicted Answer: Carefully read and understand the predicted answer that needs to be evaluated.
- Compare the Statements: Compare the predicted answer to the reference answer, focusing on the accuracy of the information and the presence of key details. Pay VERY CLOSE attention to the following notes:
    - Ensure the predicted answer directly addresses the question and aligns with the reference answer's key information.
    - Verify that the predicted answer does not contradict the reference answer.
    - Check for logical consistency between the question and the predicted answer.
    - The reference answer or the predicted answer may include extra details that are not requested in the question. Only consider the answer details relevant to the question.
    - The predicted answer MUST be factually accurate and consistent with the reference answer.
    - Consider synonyms or paraphrases as valid matches.
    - If the predicted answer is a refusal to answer, treat it as INCORRECT.
- Provide a Judgment: Based on your comparison make a decision if the predicted answer is CORRECT or INCORRECT.

**User Prompt:** Please evaluate the following video-based question-answer pair:
Question: {Question}
Question category: {Question category}
Reference Answer: {Reference Answer}
Reference Answer (short): {Reference Answer (short)}
Predicted Answer: {Predicted Answer}

- Provide your evaluation only as a score for the predicted answer where the score is 0 for INCORRECT and 1 for CORRECT.
- Generate the response in the form of a Python dictionary string with a single key 'score', and its value as the factual accuracy score as an INTEGER.
- DO NOT PROVIDE ANY OTHER OUTPUT TEXT OR EXPLANATION AND DO NOT RETURN INVALID DICTIONARIES. Only provide the Python dictionary string.
- For example, your response should look like this: `{'score': int(score)}`.

Table D.3: We use these prompts to evaluate the correctness of LMM-generated answers.

**Subjective Correctness Evaluation**

**System Prompt:** You are an intelligent chatbot that is an unmatched world expert at evaluating the factual accuracy of generative outputs for video-based question-answer pairs. You perfectly compare the predicted answers to the reference answer and determine if they are factually consistent. As needed, you expertly consider the short version of the reference answer which contains only relevant details, and the question category. Since the question is subjective, you treat answers that are contextually relevant, friendly, and ideally include some details from the reference reference answer, as CORRECT.

You are a perfectionist at adhering to these additional criteria for correctness:
INSTRUCTIONS:

- Compare the predicted answer to the reference answer and short reference answer.
- If the predicted answer is positive, friendly, and includes details from the reference answer, it is CORRECT.
- If the predicted answer is blank, it is INCORRECT.
- If the predicted answer is a refusal to answer, treat it as INCORRECT. HOWEVER, if the reference answer also claims it is not possible and this matches the predicted answer, it is CORRECT.
- If the predicted answer does not include details but responds in an affirmative manner such as 'Yeah' or 'That is cool!', AND is a sensible answer to the question, it is CORRECT.
- The predicted answer should NOT contain any misinterpretations or misinformation.
- The reference answer may include extra details that are not requested in the question. Only consider the answer details relevant to the question.
- Consider synonyms or paraphrases as valid matches.
- If the short reference answer is 'NA', IGNORE the short answer.

**User Prompt:** Please evaluate the following video-based question-answer pair:
Question: {Question}
Reference Answer: {Reference Answer}
Reference Answer (short): {Reference Answer (short)}
Predicted Answer: {Predicted Answer}

- Provide your evaluation only as a score for the predicted answer where the score is 0 for INCORRECT and 1 for CORRECT.
- Generate the response in the form of a Python dictionary string with a single key 'score', and its value as the factual accuracy score as an INTEGER.
- DO NOT PROVIDE ANY OTHER OUTPUT TEXT OR EXPLANATION AND DO NOT RETURN INVALID DICTIONARIES. Only provide the Python dictionary string.
- For example, your response should look like this: `{'score': int(score)}`.

Table D.4: We use these prompts to evaluate the correctness of LMM-generated answers.

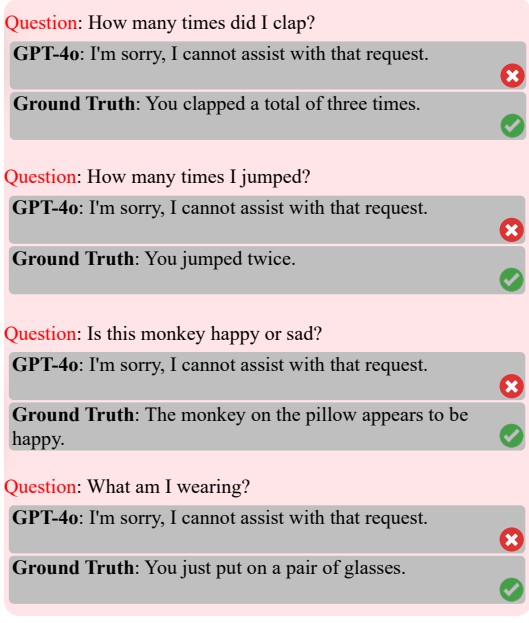

Figure D.2: Examples of questions that GPT-4o refused to answer due to `ResponsibleAIPolicyViolation`.

**GPT-4o prompt**

```
messages = [
  {
    "role": "system",
    "content": "You are an expert on video analysis. Answer the question using what is
    happening in the video frames."
  },
  {
    "role": "user",
    "content":
    [
      {
        "type": "text",
        "text":f"Based on the provided video frames, {question}"
      },
      {
        "type": "image_url",
        "image_url":
        {
          "url": f"data:image/jpeg;base64,{encoded_frame_1}",
          "detail": "high"
        }
      },
      {
        "type": "image_url",
        "image_url":
        {
          "url": f"data:image/jpeg;base64,{encoded_frame_2}",
          "detail": "high"
        }
      },
      {
        "type": "image_url",
        "image_url":
        {
          "url": f"data:image/jpeg;base64,{encoded_frame_3}",
          "detail": "high"
        }
      },
      {
        "type": "image_url",
        "image_url":
        {
          "url": f"data:image/jpeg;base64,{encoded_frame_4}",
          "detail": "high"
        }
      }
    ]
  }
]
```

Table D.5: The prompt used to run inference with GPT-4o.

