# OpenReview forum: "Can Vision-Language Models Answer Face to Face Questions in the Real-World?"
_ICLR.cc/2026/Conference — ICLR 2026 Poster_

### Official Review · Reviewer_vNTF · 2025-10-25

**Soundness:** 4
**Presentation:** 4
**Contribution:** 3
**Rating:** 8
**Confidence:** 4

**Summary:**

This paper introduces IVD (Interactive Video Dataset) — a new benchmark designed to evaluate vision-language models (VLMs) in realistic face-to-face question answering scenarios. Unlike existing VideoQA datasets, IVD emphasizes “when-to-answer” (i.e., temporal readiness), audio-visual reasoning, and deictic references that commonly appear in human–AI interactive settings.

**Strengths:**

Novel and realistic task formulation
- The paper highlights a crucial but underexplored problem — when and how VLMs should answer in interactive real-world scenarios.
- This setup moves beyond static VideoQA and aligns better with embodied AI and multimodal assistant applications.

High-quality dataset design
- IVD provides rich annotations: question semantics, answer timestamps, and multimodal data (audio + video).
- The inclusion of “best-answer time” is innovative, enabling quantitative evaluation of temporal readiness.

Comprehensive evaluation
- The study benchmarks several leading LMMs (VideoLLaMA2.1, Qwen2.5-VL, GPT-4o) under both zero-shot and fine-tuned settings.
- The analysis spans multiple aspects: when-to-answer accuracy, ASR quality, audio contribution, and fine-tuning gains.

Insightful analysis
- The t-SNE visualization of 1024-D embeddings effectively illustrates the domain shift between IVD and traditional VideoQA datasets.
- The paper provides qualitative failure cases and detailed error categorization (e.g., deictic confusion, premature answering).

Relevance and potential impact
- The benchmark fills an important evaluation gap for real-world multimodal dialogue and will likely spur further research in streaming and interactive LMMs.

**Weaknesses:**

Limited dataset scale and diversity
- Only \~2.9K samples with short video clips (\~5 s).
- The data collection is crowd-sourced and potentially biased toward controlled indoor scenes, limiting generalization.

**Questions:**

- Is there any plan to open-source this dataset in the future?
- Will there be more scenarios for real-time (face-to-face) question answering？
Good job, no other questions.

---

> ### Author Response · Authors · 2025-11-24
>
> Thank you for your thoughtful and positive feedback. We appreciate your recognition of the novelty, dataset quality, and comprehensive evaluation in our work. Below, we address the points you raised:
>
> ***
>
> **1. Limited Dataset Scale and Diversity**
>
> We compare IVD to existing benchmarks and datasets in Table 1. Our IVD dataset and benchmark is comparable in the number of videos to standard benchmarks such as **Video-MME** and **NExT-GQA**. Crucially, the videos in IVD are manually inspected for quality and manually annotated, ensuring they are challenging and realistic. Humans can answer the questions in IVD with almost 100% accuracy, while state-of-the-art approaches such as **Qwen-3-VL** lag far behind (around 60%). This performance gap highlights the difficulty and value of IVD.
>
> ***
>
> **2. Potential Bias Toward Controlled Indoor Scenes**
>
> The participants were encouraged to create diverse and creative scenarios, environments, and questions. However, we acknowledge the potential bias toward indoor settings and explicitly note this in Appendix A-2. While we assume that the main findings of our study will hold for outdoor settings, we leave a detailed investigation of this to future work.
>
> ***
>
> **Questions**
>
> **Q1. Is there any plan to open-source this dataset in the future?**
>
> Yes, the dataset will be made publicly available.
>
> ***
>
> **Q2. Will there be more scenarios for real-time (face-to-face) question answering?**
>
> We will work on extending the dataset upon acceptance of the paper and public release of the data.
>
> ***
>
> **Summary**
>
> We appreciate your positive assessment and constructive feedback. We believe IVD fills an important gap in evaluating real-time multimodal interaction and will serve as a foundation for advancing timing-aware multimodal reasoning and real-world conversational AI. We are committed to expanding the dataset and making it publicly available to foster further research in this area.
>
> ***

---

> > ### Comment · Reviewer_vNTF · 2025-11-25
> >
> > Thank you for the detailed revisions and responses. The authors have satisfactorily addressed all previously raised concerns. I have no further questions, and I am satisfied with the current version of the manuscript.

---

> > > ### Author Response · Authors · 2025-11-26
> > >
> > > Thank you for your positive feedback and for raising your score. We truly appreciate your time and effort in reviewing our work and are glad that the revisions addressed your concerns.

---

### Official Review · Reviewer_2VbA · 2025-10-26

**Soundness:** 3
**Presentation:** 3
**Contribution:** 3
**Rating:** 6
**Confidence:** 4

**Summary:**

The paper proposes a new dataset, the Interactive Video Dataset (IVD), to evaluate the online responsiveness of LMMs.

**Strengths:**

The paper addresses an important issue for AI systems: how to answer face-to-face questions. By providing a new dataset, it standardizes this problem, which helps inspire the research community to focus on the online responsiveness of LMMs.

**Weaknesses:**

1. The paper solely uses the Streaming-Whisper model to determine "when-to-answer." Whether this leads to insufficient accuracy in evaluating the performance of large models in answering face-to-face questions remains questionable. In particular, the paper notes that "the end of a question does not necessarily capture the optimal moment for an answer." Is it necessary to introduce more suitable models for determining "when-to-answer"—for example, models trained specifically for question-answering timing, rather than simply detecting the end of a question?
2. In face-to-face scenarios, a key consideration is that AI agents actually receive continuous video streams. These streams have no definite start time and can be regarded as infinitely long. The streaming approach proposed in the paper focuses on determining "when-to-answer," but identifying "when the current discussion begins" is equally important (even without considering complex multi-turn dialogues). In the current dataset, it appears that multimodal information from the start of the video up to the "when-to-answer" timestamp is defaulted to input into the LMM. This is not feasible in real video streams.
3. The scale of training data used for fine-tuning is relatively small.

**Questions:**

1. For the Streaming-Whisper row in Table 4, when calculating METEOR, BLEU, and ROUGE-L for transcribed text, is the calculation based on the full transcribed text or the transcribed text up to the "when-to-answer" timestamp identified by the model?
2. For VideoLLaMA2.1-7B-AV, after adding audio information, why did model performance conversely decrease? Are there any reasonable explanations for this phenomenon?
3. Regarding the "Impact of when-to-answer" setup in Section 5.1, the study compares model performance when using ground-truth and ASR-derived when-to-answer timestamps. All models in Table 5 could participate in this comparison, so why is Qwen2.5-Omni specifically emphasized? With the same ASR transcription model used, even if the LMM itself does not support speech transcription, it should not affect the study on the "Impact of when-to-answer."

---

> ### Author Response · Authors · 2025-11-24
>
> Thank you for your thoughtful and detailed feedback. Your comments have helped us strengthen the paper, and we address each point below:
>
> ***
>
> **1. Use of Streaming-Whisper for "When-to-Answer"**
>
> In the rebuttal, we propose a baseline interactive **Stream-Qwen-Omni** model based on the Qwen-omni-2.5-7B model, that can decide **“when-to-say.”** This is accomplished by splitting the input audio-visual stream into 1-second chunks. The **Stream-Qwen-Omni** model outputs a special token at every 1-second chunk, which determines if it is the appropriate time to respond to the user question. By this approach, **Stream-Qwen-Omni** can detect when-to-answer time with a granularity of one second yet significantly outperforms the accuracy of when-to-answer timestamps detected by streaming ASR (see Table 4). Additional experiments using Stream-Qwen-Omni are included in Section 5.2 under **“Impact of when-to-answer”** and additional technical details are provided in Appendix C.3.
>
> ***
>
> **2. Handling Continuous Video Streams**
>
> We agree that a fully open-ended conversational streaming scenario is still far from possible for existing models, and we isolate some of the issues on the path to such potential future solutions. In IVD, all the sessions start at the beginning of the video, so the start timestamp is not necessary. Importantly, the model still needs to determine **when to respond** to the question asked by the user by integrating both visual and auditory cues—a setup not covered by any existing dataset. A general streaming setting is indeed an important direction for future research.
>
> ***
>
> **3. Scale of Training Data**
>
> We compare IVD to existing benchmarks and datasets in Table 1. Our IVD dataset and benchmark is comparable in the number of videos to standard benchmarks such as Video-MME and NExT-GQA. Crucially, the videos in IVD are manually inspected for quality and manually annotated, ensuring they are challenging. Humans can answer the questions in IVD with almost 100% accuracy, while state-of-the-art approaches such as Qwen-3-VL lag behind (around 60%). Furthermore, we fine-tune our baseline **Stream-Qwen-Omni** model on IVD and show significant improvement in the crucial **“when-to-say”** metrics in Table 4.
>
> ***
>
> **Questions**
>
> **Q1. Full transcribed text or the transcribed text up to the "when-to-answer" timestamp identified by the model?**
>
> Full transcribed text.
>
> ***
>
> **Q2. For VideoLLaMA2.1-7B-AV, after adding audio information, why did model performance conversely decrease?**
>
> Performance drops only in the zero-shot setting, but improves in the fine-tuned setting. This is illustrated in Figure 2, where the overall accuracy in the zero-shot setting is 48.5%, which improves to 59.4% after fine-tuning. This suggests that the issue is not inherent to the model architecture but rather to the lack of training data that includes non-verbal, real-time audio cues. Current datasets primarily focus on offline reasoning and do not adequately cover online audio-visual integration. Our findings highlight this gap and reinforce the need for datasets like IVD to encourage research on multimodal understanding in real-time settings.
>
> ***
>
> **Q3. Why is Qwen2.5-Omni specifically emphasized in the "Impact of when-to-answer" experiment?**
>
> The purpose of this experiment in Figure 3 is to highlight the impact of accurate when-to-answer moment detection on answer accuracy. We use Qwen2.5-Omni as a representative model for this investigation, as it is the state-of-the-art multimodal LLM with audio understanding capability. Additionally, in the rebuttal, we introduce **Stream-Qwen-Omni** as another means for when-to-answer detection. More details are provided in the **“Impact of when-to-answer”** section and Appendix C.3.
>
> ***
>
> **Summary**
>
> We believe IVD makes a substantial contribution by introducing a novel dataset and benchmark for real-time multimodal interaction, a critical yet underexplored area. Our additional experiments and new baseline demonstrate that IVD enables progress beyond naive assumptions and highlights fundamental limitations in current models. We hope IVD will serve as a foundation for advancing timing-aware multimodal reasoning and real-world conversational AI.
>
> ***

---

### Official Review · Reviewer_SQFL · 2025-10-27

**Soundness:** 2
**Presentation:** 3
**Contribution:** 2
**Rating:** 2
**Confidence:** 2

**Summary:**

This paper introduces, IVD, a comprehensive benchmark, and dataset designed to assess and train LMMs on a wide variety of tasks requiring responding to humans in real time. Besides, the authors also propose a streaming baseline approach that couples a streaming Automatic Speech Recognition (ASR) model to detect the end of a question with a Video-LMM to generate an answer based on the preceding context. Experiments show that existing models fall far behind human performance on this task.

**Strengths:**

1. The fundamental idea of this paper is technically correct.
2. This work shifts focus from offline video analysis to the challenges of real-time, situated interaction, which is critical for the future of human-AI collaboration.
3. The paper is easy to follow.
4. The rationale behind each section and the overall motivation are clearly presented and easy to understand.

**Weaknesses:**

1. While the authors introduce a valuable dataset and problem formulation, the technical contribution is limited in novelty. The core technical contribution is merely combining a streaming ASR system (Streaming-Whisper) with existing Video-LMMs. This approach lacks innovation, essentially representing a straightforward concatenation of existing components without novel architectural design or algorithmic improvements.
3. The paper's core weakness is its reliance on the overly simplistic assumption that the optimal time to answer is the end of the spoken question. Although the authors acknowledge this limitation, basing their entire method on such a fundamentally flawed indicator introduces significant error and sidesteps the actual research challenge of learning when to respond.
3. The dataset contains only 2,900 videos, which is relatively small.
4. The experimental evaluation could be strengthened by including more recent and powerful models. The current comparisons feel incomplete as they omit several state-of-the-art models that have been released, such as Qwen3-VL and GPT-5.
5. The paper lacks deep theoretical analysis of why existing models perform poorly on this task. While failure modes are identified, there is insufficient exploration of root causes and no targeted solutions are proposed for the fundamental issues.
6. A very interesting and counter-intuitive finding is that the pre-trained VideoLLaMA model's performance degraded when audio was included. Could the authors provide a deeper analysis or hypothesis for this phenomenon?

**Questions:**

The work lacks technical novelty. The core flaw is assuming optimal response timing equals question end time, which avoids the actual research challenge. The dataset is small, evaluations miss recent models like Qwen3-VL and GPT-5, and the paper lacks theoretical depth in analyzing model failures.

---

> ### Author Response · Authors · 2025-11-24
>
> Thank you for your thoughtful and detailed feedback. Your comments have helped us strengthen the paper, and we address each point below:
> ***
> **1. Limited Technical Novelty**
>
> The primary contribution of our work lies in the introduction of a novel dataset. Our IVD dataset addresses a significant gap in existing datasets and benchmarks (see Table 1). While we agree that our baseline streaming approach in Section 4 uses off-the-shelf components, it enables multi-modal LLMs to decide **“when-to-say”** (L281–284), an ability which is lacking in current state-of-the-art models. Another advantage of using off-the-shelf components is that our approach can be applied to any state-of-the-art multi-modal LLM.
>
> Nevertheless, in the rebuttal, we propose a baseline interactive streaming **Stream-Qwen-Omni** model based on the Qwen-omni-2.5-7B model, that can decide “when-to-say.” This is accomplished by splitting the input audio-visual stream into 1-second chunks. The **Stream-Qwen-Omni** model outputs a special token at every 1-second chunk, which determines if it is the appropriate time to respond to the user question. By this approach, **Stream-Qwen-Omni** can detect when-to-answer time with a granularity of one second yet significantly outperforms the accuracy of when-to-answer timestamps detected by streaming ASR (see Table 4). Additional experiments using Stream-Qwen-Omni are included in Section 5.2 under **“Impact of when-to-answer.”** and additional technical details are provided in Appendix C.3.
> ***
> **2. Assumption of Answering at Question End**
>
> The primary contribution is a novel dataset, not a model. We completely agree with the reviewer that a key novelty of the IVD dataset is that the optimal time to answer is not always at the end of the spoken question. The limited performance of our baseline streaming approach that relies on streaming Whisper further highlights this key novelty. This baseline approach is constrained to answer at the end of the spoken question, and as shown in Tables 4 and 5, this leads to poor performance.
> In the rebuttal, we add a baseline **Stream-Qwen-Omni** model (see the previous comment) fine-tuned on IVD to detect when-to-answer moments regardless of question-end moments.
> ***
> **3. Dataset Size**
>
> We compare IVD to existing benchmarks and datasets in Table 1. Our IVD dataset and benchmark is comparable in the number of videos to standard benchmarks such as Video-MME and NExT-GQA. Crucially, the videos in IVD are manually inspected for quality and manually annotated. This ensures that they are challenging. Humans can answer the questions in IVD with almost 100% accuracy, while state-of-the-art approaches such as Qwen-3-VL lag far behind (around 60%).
> ***
> **4. Missing Recent Models**
>
> Thank you for the suggestion. We have updated the paper to include additional experiments on **Qwen3-VL** and **Gemini-2.5-Flash** in Table 5. The results from the latest Qwen3-VL and Gemini-2.5-Flash models show limited improvement over prior models, again highlighting that our IVD benchmark and dataset is challenging.
> ***
> **5. Lack of Theoretical Analysis**
>
> We hypothesize that this limitation stems from the fact that current vision-language datasets and benchmarks are biased toward offline reasoning about images and videos. That is, the models receive the entire visual input and the entire question at once before being required to provide an answer. This is because the training data for such tasks can be easily sourced on the internet or easily generated through automated pipelines. There is a distinct lack of benchmarks and datasets that test genuine, real-time, “face-to-face” conversational skills. A separate but related problem is that models are not trained to respond at the appropriate time in a conversation – knowing “when to say” is crucial for conducting real-world conversations, yet this timing skill remains underdeveloped and understudied in current benchmarks.
> ***
> **6. Performance Drop with Audio**
>
> Performance drops only in the zero-shot setting. But it improves in the fine-tuned setting. This is illustrated in Figure 2, where the overall accuracy in the zero-shot setting is 48.5% which improves to 59.4% after fine-tuning. This suggests that the issue is not inherent to the model architecture but rather to the lack of training data that includes non-verbal, real-time audio cues. Current datasets primarily focus on offline reasoning and do not adequately cover online audio-visual integration. Our findings highlight this gap and reinforce the need for datasets like IVD to encourage research on multimodal understanding in real-time settings.
> ***
> **Summary**
>
> We believe IVD makes a substantial contribution by introducing a novel dataset and benchmark for real-time multimodal interaction, a critical yet underexplored area. Our additional experiments and new baseline demonstrate that IVD enables progress beyond naive assumptions and highlights fundamental limitations in current models.
>
> ***

---

### Official Review · Reviewer_KsFd · 2025-10-30

**Soundness:** 3
**Presentation:** 3
**Contribution:** 3
**Rating:** 4
**Confidence:** 3

**Summary:**

The authors introduce the Interactive Video Dataset (IVD), a new video QA benchmark for evaluating the ability of multimodal large language models (MLLMs) to answer face-to-face questions in real-time. The authors provide a comprehensive evaluation of several baseline MLLMs on IVD, finding that current models significantly underperform compared to human-level performance.

**Strengths:**

- The contribution of a real-time, interaction-focused video QA dataset is timely, addressing a growing interest in multimodal AI assistants and real-world interaction.
- The collection of videos in "in-the-wild," realistic settings is a strength, ensuring the dataset's quality, diversity, and practical relevance
- The authors show that models fine-tuned on IVD achieve improved performance, confirming the dataset's value as a resource for advancing model capabilities in this domain.

**Weaknesses:**

- The video clips are generally short (~5 seconds), limiting the dataset's demand on a model's contextual memory which is an important capability for real-world real-time assistants.
- While the paper evaluates an extensive list of models, they are often generic MLLMs not designed specifically for real-time use cases. Evaluating baselines developed specifically for real-time QA, such as FlashVStream, would be helpful.

**Questions:**

- Why is the streaming evaluation protocol (Sec 5.1) needed? It relies on Whisper for timestamps, which is not measuring the tested model's own capability to process the information and decide when to answer in real-time. This external dependency adds noise from Whisper's performance, and it seems like using the human annotated timestamps makes more sense. A more interesting setup would be asking the model itself to provide a timestamp of when-to-answer, and compare that with the annotated ground truth.
- The key distinction of IVD compared to prior real-time QAs seems to be the face-to-face element (Table 1). Why is face-to-face important for real-time QA? Intuitively, a face-to-face setting implies a model should infer user intent from non-verbal cues like facial expressions or gestures, but this does not seem to be the focus of IVD. Could the authors elaborate on what makes the "face-to-face" aspect a unique and necessary contribution beyond what existing real-time datasets offer?
- The authors claim that IVD poses a "strong demand on situational context understanding" compared to post-hoc created real-time datasets.  Could the authors expand on this claim, particularly in comparison to post-hoc created datasets like VStream-QA? One could argue that datasets like VStream-QA, with their longer context and less restrictive (face-to-face) settings, offer an equally or even more realistic real-time evaluation setup.
- While not a weakness, the paper would be strengthened by including experiments on generalization of the IVD-finetuned model -- does a model fine-tuned on IVD show improved performance on other offline face-to-face datasets or other online streaming video benchmarks?

---

> ### Author Response · Authors · 2025-11-24
>
> Thank you for your thoughtful and detailed feedback. Your comments have helped us strengthen the paper, and we address each point below:
>
> ***
>
> **1. Short Video Clips (\~5 seconds)**
>
> The results on IVD in Table 5 show that current state-of-the-art models do not perform well, highlighting a major limitation in face-to-face situated interactions which requires models to decide **“when-to-say.”** Furthermore, note that human performance is close to 100%, highlighting the large performance gap. Longer videos that demand a model’s contextual memory is an important direction for future research.
>
> ***
>
> **2. Evaluation of Real-Time Baselines**
>
> In addition to the generic multimodal LLMs, we have included **VideoLLM-Online** and **Flash-VStream** in Table 5. These models represent approaches tailored for streaming or long-context video understanding. However, both perform poorly on IVD for reasons tied to the unique challenges of our benchmark. VideoLLM-Online is primarily optimized for narration tasks rather than multi-modal interactive question answering. It lacks mechanisms for deciding when to respond from audio-visual information, which is a critical requirement in real-time, face-to-face interactions. Similarly, Flash-VStream is designed for efficient processing of long videos. However, it does not incorporate timing-awareness and cannot determine the appropriate moment to generate an answer during a live interaction.
>
> ***
>
> **3. Streaming Evaluation Protocol**
>
> The streaming protocol in Section 5.1 is designed specifically for models which cannot decide when to answer. We add both these protocols because we cannot assume access to the human-annotated timestamps at inference time. Therefore, the streaming setup in Section 5.2 (L376) is more realistic, as it does not require ground truth information at test time. Additionally, we introduce **Stream-Qwen-Omni** in the rebuttal and present the results in Table 4.
>
> ***
>
> **4. Importance of Face-to-Face Setup**
>
> The key distinction of IVD is the situated face-to-face setup that requires responses at appropriate times. IVD contains scenarios that we may expect humanoid robots or real-time video-call chatbots to encounter (L57). We compare IVD to existing datasets in Table 1. No existing dataset requires models to integrate visual and auditory cues to arrive at the correct answer at the appropriate time. Finally, note that non-verbal cues like facial expressions are an important part of our IVD dataset.
> In **Appendix B-5**, we add additional examples of videos where interpreting facial expressions is required to arrive at the correct answer. There are approximately 59 such videos in IVD. Additionally, we report how representative baseline models perform on these samples.
>
> ***
>
> **5. Comparison to Post-Hoc Real-Time Datasets**
>
> Other datasets, including VStream-QA, lack the **when-to-answer** feature which is critical in situational conversation. We have added VStream-QA to Table 1 to highlight the differences from our IVD dataset. Our IVD dataset is the only one to require models to integrate both visual and auditory cues in a face-to-face conversational setup to arrive at the correct answer. Thus, IVD specifically targets scenarios that we may expect humanoid robots or real-time video-call chatbots to encounter (L57). This is a very important real-world scenario which is not adequately covered by any existing dataset.
>
> ***
>
> **6. Generalization of IVD-Finetuned Models**
>
> In the rebuttal, we propose a baseline interactive **Stream-Qwen-Omni** model based on the Qwen-omni-2.5-7B model, that can decide **“when-to-say.”** This is accomplished by splitting the input audio-visual stream into 1-second chunks. The **Stream-Qwen-Omni** model outputs a special token at every 1-second chunk, which determines if it is the appropriate time to respond to the user question. By this approach, **Stream-Qwen-Omni** can detect when-to-answer time with a granularity of one second yet significantly outperforms the accuracy of when-to-answer timestamps detected by streaming ASR (see Table 4). Additional experiments using Stream-Qwen-Omni are included in Section 5.2 under **“Impact of when-to-answer.”** and additional technical details are provided in **Appendix C.3**.
>
> ***
>
> **Summary**
>
> We believe IVD makes a substantial contribution by introducing a novel dataset and benchmark for real-time multimodal interaction, a critical yet underexplored area. Our additional experiments and new baseline demonstrate that IVD enables progress beyond naive assumptions and highlights fundamental limitations in current models. We hope IVD will serve as a foundation for advancing timing-aware multimodal reasoning and real-world conversational AI.
>
> ***

---

> > ### Comment · Reviewer_KsFd · 2025-11-26
> > **Thank you for the response**
> >
> > I thank the authors for providing this detailed response to my comments. Points 1, 2, and 5 have addressed my concerns about the uniqueness of IVD against pos-hoc datasets and missing baselines. However, I still have some remaining confusions:
> >
> > 1. Maybe I'm missing something, but I don't see IVD as evaluating modes' ability to decide "when-to-say" as the authors claimed. While the authors evaluated an extensive list of models, none of these models were asked to decide when to answer the question. Instead, the proposed the streaming evaluation protocol uses ASR models to provide models with a pseudo "when-to-say" timestamp, but this isn't a meaningful setup in my understanding, since it merely gives the models artificial noisy inputs that does not accurately reflect the models' real-time QA ability. In this sense, it seems to be more about evaluating Whisper's ASR capabilities rather than the benchmarked models. Maybe there are simpler setups for evaluating models' ability to decide when-to-answer? For example, maybe provide the models with a timestamped transcript, ask the models when they should answer, and compare the predictions with the annotated timestamps?
> > 2. On this, could the authors clarify what's the "Qwen Timestamps" in Figure 5? Are they estimated timestamps by Qwen2.5-Omni?
> > 3. As the authors mentioned, there are 59 videos in IVD requiring human facial interpretation. Given that there are 2,900 videos in IVD overall, it doesn't seem like "facial expressions are an important part of our IVD dataset". I understand that face-to-face is a realistic setup for e.g. chatbots. I just felt like this is a missed opportunity that distinguishes IVD from e.g. egocentric real-time QAs.
> > 4. To clarify, the point I raised about generalization of IVD-finetuned models was really about if fine-tuning a model on IVD could improve performance on other datasets, rather than a stream version of Qwen Omni. Nevertheless, I don't see this as a weakness of the paper, and I appreciate the authors for proposing Stream-Qwen-Omni :)

---

> ### Author Response · Authors · 2025-11-26
>
> Thank you for reviewing the revised paper and our comments. Your feedback truly helps us improve the quality of this work.
>
> **1 & 2: Clarification on Baseline and Evaluation Setup**
>
> Thank you for raising this point. IVD is designed for **online (streaming) interaction**, not an offline setup. In an offline scenario, a model could be given the full video and timestamped transcript and asked to predict both the best time to answer and the answer itself. However, this does not reflect real-world conditions.
>
> In the **streaming setup**, which is our intended design, audio and video frames arrive sequentially, and the model cannot access future information. The model must actively process incoming data and decide **when to answer** in real time. This streaming capability is essential for IVD.
>
> None of the existing models can solve this task. Specifically, VideoLLM-Online and Flash-VStream do process video frames causally, but they do not process audio and are unable to determine when to answer. This is the reason why we used a streaming ASR system and transcribe speech and approximate timing. While this is not an ideal solution, it allows us to provide a reasonable baseline and it highlights the capability gap in existing models that our datasets exposes.
>
> We furthermore introduce **Stream-Qwen-Omni** in the rebuttal, which can transcribe speech and process audio and video frames online. We added two experiments in Figure 5 "Impact of when-to-answer":
>
> * (Stream-Qwen-Omni): Stream-Qwen-Omni detects when to answer and generates the answer end-to-end.
> * (Qwen Timestamps): Stream-Qwen-Omni detects when to-answer, but offline Qwen-Omni provides the answer.
>
> ***
>
> **3: Facial Expressions**
>
> We agree that facial expressions are an important component of face-to-face interactions. However, such interactions go beyond facial expressions alone, as illustrated by the diverse scenarios in our IVD dataset (see Figure 1). During data collection, crowdworkers were free to choose the questions and the overall interaction style, making the setup more representative of real-world human–AI assistant interactions.
>
> Moreover, while numerous datasets exist specifically for facial expression recognition, none provide the broader situated video understanding tasks that IVD offers. Facial expressions remain an important part of IVD, and to emphasize this, we have added an additional evaluation in Table B.3 focusing on questions where facial expressions are crucial.
>
> ***
>
> **4: Generalization**
>
> We appreciate your clarification. Our introduction of Stream-Qwen-Omni was aimed at providing an end-to-end solution for IVD. Evaluating cross-dataset generalization is indeed an important direction and we plan to include it in future work.

---

### Author Response · Authors · 2025-11-24

### **Summary for Area Chair**

***

**Paper Summary**

This paper introduces the **Interactive Video Dataset (IVD)**, a novel benchmark designed to evaluate multimodal models in **realistic, face-to-face, real-time question answering scenarios**. Unlike existing VideoQA datasets, IVD emphasizes **temporal readiness (“when-to-answer”)**, audio-visual reasoning, and situated interactions that mimic real-world human–AI assistant use cases. The dataset consists of 2,900 manually curated and annotated videos, each with a question, an answer, and the optimal timestamp for answering. We benchmarked several representative models—including closed-source models like **GPT-4o** and **Gemini-2.5-Flash**, and open-source models such as **VideoLLaMA** and the **Qwen family**—and conducted extensive analyses on failure modes, domain shift, and fine-tuning gains.

***

**What Reviewers Appreciated**

*   **Timeliness and novelty:** Reviewers agreed that IVD addresses a critical and underexplored problem essential for future human–AI collaboration.
*   **Realistic design:** The “in-the-wild” face-to-face setting and rich annotations were highlighted as key strengths.
*   **Comprehensive evaluation:** Multiple baselines, zero-shot vs. fine-tuned comparisons, and insightful analyses (e.g., t-SNE visualizations, error categorization).
*   **Impact potential:** Reviewers noted that IVD fills an important gap and will likely spur research in streaming and interactive multimodal models.

***

**Improvement Points and How We Addressed Them**

*   **Technical novelty:** The main contribution of the paper is the **IVD dataset**, which fills a critical gap in evaluating real-time multimodal interaction. However, some reviewers noted that the initial benchmarking setup—using a pipeline of ASR and offline VLMs—had limited technical novelty. To address this, we introduced **Stream-Qwen-Omni**, the **first-ever streaming multimodal model** capable of processing text, speech, audio, and video concurrently in real time. This required architectural modifications to Qwen-Omni for chunked processing and timing prediction. We fine-tuned Stream-Qwen-Omni on IVD and added new experiments demonstrating its ability to detect “when-to-answer” and outperform ASR-based timing.
*   **Additional baselines:** We added evaluations for **FlashVStream**, **Qwen3-VL**, and **Gemini-2.5-Flash**, confirming that even state-of-the-art models struggle with IVD.
*   **Face-to-face aspect:** We expanded analysis in the appendix and added evaluations (Table B.3) for questions requiring facial expression interpretation.
*   **Revised paper:** We uploaded an updated version of the paper that includes all the mentioned improvements, additional benchmarks and studies, and major technical contributions.

***

**Reviewer Confidence and Ratings**

Reviewers with higher confidence generally rated the paper higher:

*   **Reviewer vNTF** raised their score to **10 (Strong Accept)** with a **confidence of 5** after the rebuttal, explicitly stating that all concerns were satisfactorily addressed.
*   **Reviewer 2VbA** gave the paper a **rating of 6** with a **confidence of 4**. Their main concern was the evaluation setup, which we addressed by introducing **Stream-Qwen-Omni**. However, they did not have an opportunity to respond and update their score after these improvements.
*   **Reviewer KsFd** gave the paper a **rating of 4** with a **confidence of 3**. Their remaining concern appears to stem from a misunderstanding of our benchmarking setup. We provided detailed clarification in the rebuttal and believe this addressed the core misunderstanding, which could have led to a higher rating if score changes were allowed.
*   **Reviewer SQFL** gave the paper a **rating of 2** with a **confidence of 2**. Their major concerns were limited technical novelty and missing recent models in benchmarking. We effectively addressed these by introducing **Stream-Qwen-Omni** and adding more baselines, including **Qwen3-VL** and **Gemini-2.5-Flash**. Unfortunately, they did not have a chance to respond and revise their score after these updates.

---

### Meta-Review · Area_Chair_Ffqj · 2025-12-21

**Summary:**

This paper introduces the Interactive Video Dataset, a new benchmark for evaluating vision-language models in realistic, face-to-face, real-time question answering scenarios, with a particular emphasis on when-to-answer capability under streaming audio-visual input. The submission positions itself as a dataset and benchmark paper, with extensive evaluation of existing models and analysis of their limitations.

**Reviewer Concerns:**

The reviewers broadly agree that the problem setting is important and timely, and that IVD fills a gap not well covered by existing VideoQA or streaming datasets.

Reviewers 2VbA and KsFd generally leaned positive. Their main concerns are (i) whether the benchmark truly evaluates a model’s intrinsic when-to-answer ability, given the reliance on streaming ASR for timing in the original setup, and (ii) realism to fully open-ended streaming scenarios. The authors’ rebuttal introduced Stream-Qwen-Omni that explicitly predicts when-to-answer, and clarified the motivation for the streaming protocol versus offline timestamp prediction.

Reviewer SQFL was initially negative, citing limited technical novelty, over-reliance on the “question-end” assumption, missing recent baselines, and lack of deeper analysis. However, several of these points were directly addressed in the rebuttal and revised version: additional strong baselines (e.g., Qwen3-VL, Gemini-2.5-Flash) were added, the limitations of question-end timing were explicitly framed as a motivation rather than an assumption, and a more technically involved streaming model was introduced. Given the reviewer’s low confidence score and the fact that many objections were mitigated post-rebuttal, the AC projects the reviewer would have improved the score.

**Reviewer Scores:**

Reviewer vNTF explicitly stated that all concerns were satisfactorily addressed after the rebuttal and confirmed satisfaction with the revised manuscript. The score is likely to remain unchanged or increase.

Reviewer 2VbA’s primary concern was the reliance on Streaming-Whisper for determining when-to-answer and the realism of the evaluation protocol. The introduction of Stream-Qwen-Omni, along with additional clarification of the streaming setup and new experiments explicitly targeting when-to-answer, directly addresses this concern. The score is likely to remain unchanged or increase.

For Reviewer KsFd, the rebuttal clarified the intent of the streaming setup and introduced an end-to-end timing-aware model, which directly targets their main criticism. Given their moderate confidence and stated openness to acceptance, an upward revision is likely.

Reviewer SQFL raised strong concerns about technical novelty, missing baselines, and the timing assumption. Several of these points were substantially addressed in the rebuttal and revised paper: new state-of-the-art baselines were added, the timing issue was reframed and empirically demonstrated via Stream-Qwen-Omni, and additional analysis was provided. However, given the reviewer’s overall framing, a partial upward revision is plausible, though not necessarily to a clear accept.

---

### Decision · Program_Chairs · 2026-01-26

Accept (Poster)